# An unusual glycerol-3-phosphate dehydrogenase in *Sulfolobus acidocaldarius* elucidates the diversity of glycerol metabolism across Archaea
Christian Schmerling[1,9], Carsten Schroeder[1,9], Xiaoxiao Zhou[1,9], Jan Bost[2], Bianca Waßmer[2], Sabrina Ninck [3], Tobias Busche[4], Lidia Montero[5,6,8], Farnusch Kaschani[3,7], Oliver J. Schmitz[5,6], Jörn Kalinowski[4], Markus Kaiser [3], Sonja-Verena Albers [2], Christopher Bräsen [1] ✉ & Bettina Siebers [1] ✉

Glycerol is highly abundant in natural ecosystems and serves as both an important carbon source for microorganisms as well as a promising feedstock for industrial applications. However, the pathways involved in glycerol degradation in Archaea remain unclear. Here, we show that the thermoacidophilic Crenarchaeon *Sulfolobus acidocaldarius* can grow with glycerol as its sole carbon source and characterize the mechanisms involved in glycerol utilization. We show that after uptake involving facilitated diffusion, glycerol is phosphorylated to glycerol-3-phosphate by glycerol kinase (GK), followed by oxidation to dihydroxyacetone phosphate catalyzed by an unusual glycerol-3-phosphate dehydrogenase (G3PDH) with a previously undescribed type of membrane anchoring via a CoxG-like protein. Furthermore, we show that while *S. acidocaldarius* has two paralogous GK/G3PDH copies (*saci_1117-1119*, *saci_2031-2033*) with similar biochemical activity, only *saci_2031-2033* is highly upregulated and essential on glycerol, suggesting that distinct enzyme pairs may be regulated by different environmental conditions. Finally, we explore the diversity of glycerol metabolism enzymes across the Archaea domain, revealing a high versatility of G3PDHs with respect to interacting proteins, electron transfer mechanisms, and modes of membrane anchoring. Our findings help to elucidate the mechanisms involved in glycerol utilization in Archaea, highlighting unique evolutionary strategies that likely enabled adaptation to different lifestyles.

Glycerol ($C_3H_8O_3$) is a simple organic compound that is an integral constituent of membrane phospholipids and storage lipids like triglycerides, and thus is highly abundant in plant, animal and microbial cells. Accordingly, many organisms, from Bacteria and Archaea to complex Eukarya, can utilize glycerol as a carbon and energy source. Furthermore, glycerol has important physiological and ecological roles, such as serving as an osmolyte or regulating cross-species interactions between photosynthetic algae and archaea in hypersaline environments[1]. Glycerol also has many biotechnological uses, from the food and pharmaceutical industries to being increasingly regarded as an attractive feedstock to support the production of

[1]Molecular Enzyme Technology and Biochemistry (MEB), Environmental Microbiology and Biotechnology (EMB), Centre for Water and Environmental Research (CWE), Faculty of Chemistry, University of Duisburg-Essen, Essen, Germany. [2]Molecular Biology of Archaea, Institute of Biology II—Microbiology, University of Freiburg, Freiburg, Germany. [3]Chemical Biology, Center of Medical Biotechnology, Faculty of Biology, University of Duisburg-Essen, Essen, Germany. [4]Center for Biotechnology (CeBiTec), Bielefeld University, Bielefeld, Germany. [5]Applied Analytical Chemistry (AAC), University of Duisburg-Essen, Essen, Germany. [6]Teaching and Research Center for Separation (TRC), University of Duisburg-Essen, Essen, Germany. [7]Analytics Core Facility Essen (ACE), Center of Medical Biotechnology, University of Duisburg-Essen, Essen, Germany. [8]Present address: Laboratory of Foodomics, Institute of Food Science Research, CIAL, CSIC, Madrid, Spain. [9]These authors contributed equally: Christian Schmerling, Carsten Schroeder, Xiaoxiao Zhou. ✉e-mail: christopher.braesen@uni-due.de; bettina.siebers@uni-due.de

value-added molecules by recombinant microbes[2]. Therefore, elucidating the mechanisms involved in glycerol metabolism is critical to not only better understand the physiological and ecological roles mediated by glycerol but also to support improved use of this substrate in biotechnological applications.

Despite the importance of glycerol across all domains of life, the detailed pathways involved in glycerol utilization have only been explored in a few species, particularly in Bacteria and Eukarya. The initial step in glycerol utilization is its uptake across the cytoplasmic membrane, which often involves facilitated diffusion via glycerol-uptake facilitators (GUF), including aqua(glycerol)porins such as GlpF[3] (Fig. 1). Additionally, glycerol can also enter the cell via passive diffusion[4] or through alternative transporters[5–9]. Then, glycerol metabolism can follow two routes, which are differentially distributed across organisms (Fig. 1). The most prevalent pathway, predominantly employed by respiring organisms[9–11], converts glycerol into sn-glycerol-3-phosphate (G3P) via an ATP-dependent glycerol kinase (GK) (encoded by glpK)[12]. Then, G3P is oxidized to dihydroxyacetone phosphate (DHAP) by one of two membrane-bound glycerol-3-phosphate dehydrogenases (G3PDH), designated as GlpD (encoded by glpD) and GlpABC (encoded by the glpABC operon), with the simultaneous

reduction of a non-covalently bound flavin adenine dinucleotide (FAD) to $FADH_2$. From $FADH_2$ the electrons are transferred to the quinone pool of the respiratory chain[13–16]. GlpD is active under aerobic conditions, e.g. in E. coli[11,14–16], transferring electrons via ubiquinone to oxygen or nitrate, and is also the G3P oxidizing enzyme in the mitochondria of Eukarya[15]. Ubiquinone is the prevalent quinone of the respiratory chain under aerobic conditions in Bacteria and mitochondria. The GlpABC respiratory complex that is mainly known from Bacteria, is induced under anaerobic conditions and reduces menaquinone with the final acceptors nitrate or fumarate (in E. coli)[11,17]. Menaquinone is commonly used by anaerobic bacteria, and facultative anaerobes including E. coli change their quinone pool from ubiquinone under aerobic to menaquinone under anaerobic conditions. The A and B subunits of the complex form a soluble and active dimer[18] which is likely anchored to the membrane via the C subunit[17,19]. In addition to these two membrane-bound G3PDH enzymes, G3P oxidation to DHAP can also be performed by the G3P oxidase GlpO (encoded by the glpO gene), a cytosolic, soluble FAD-dependent enzyme directly reducing oxygen to $H_2O_2$, which is detoxified via peroxidase or catalase. GlpO is mainly known from aerotolerant/microaerophilic lactic acid bacteria and Mycoplasma spp[20–25]. The second route of glycerol processing is less abundant and restricted to fermentatively growing organisms including E. coli, some additional Enterobacteriaceae and a few other bacterial species[26,27]. In these organisms, glycerol is first oxidized to dihydroxyacetone (DHA) in an $NAD^+$-dependent manner by the glycerol dehydrogenase GldA (encoded by the gldA gene). Then, DHA is phosphorylated to DHAP in a phosphoenolpyruvate (PEP)-dependent manner by DhaK (encoded by the dhaK gene) or in an ATP-dependent manner by GlpK (encoded by the gene glpK in Klebsiella pneumoniae)[11,26]. The NADH derived from glycerol oxidation is reoxidized with another molecule of glycerol, and finally converted to 1,3-propanediol or 1,2-propanediol[26]. Both of these metabolic routes result in DHAP production, which is then channelled into the central metabolism and further metabolized via the lower common shunt of the Entner–Doudoroff (ED) and the Embden-Meyerhof-Parnas (EMP) pathway or utilized for gluconeogenesis. G3P serves as a building block for phospholipid and membrane synthesis in Bacteria and Eukarya.

While different routes of glycerol metabolism have been characterized in Bacteria and Eukarya, comparatively little is known in Archaea, with studies focusing primarily on Haloferax volcanii. H. volcanii utilizes glycerol via homologues of the bacterial GlpK and GlpABC, while GlpD is absent[1,28–30]. Although genes encoding putative GKs and G3PDHs have been described in representatives of other taxa[31], and some enzymes have been characterized[29,32–36], no other Archaea have so far been shown to grow with glycerol as sole carbon and energy source.

Here, we expand the characterization of glycerol metabolism in Archaea by delineating the mechanisms involved in glycerol utilization in the thermoacidophilic Crenarchaeon Sulfolobus acidocaldarius. S. acidocaldarius grows heterotrophically at temperatures around 75 °C and a pH of 2.0–3.0 with a variety of carbon sources, although like other Sulfolobales, S. acidocaldarius has been reported to not grow on glycerol[37]. S. acidocaldarius is regarded as a promising chassis for biotechnological applications, due to its ability to grow in extreme environments, availability of its genome sequence and genetic tools[38–40] and because some of its metabolic pathways, particularly those involved in carbohydrate metabolism, have been characterized in detail[38,41]. With regards to enzymes involved in glycerol metabolism, only GK homologues have been identified, while G3PDH homologues appear to be absent[31]. However, S. acidocaldarius has recently been shown to cleave triacylglycerides by esterases and to grow with fatty acids[42]. As cleavage of triacylglycerides results in the production of glycerol, here we reexamined the growth of S. acidocaldarius on this substrate. Using a combination of growth studies, genetic mutants, and biochemical and multi-omics analyses, we demonstrate that S. acidocaldarius grows using glycerol as its sole carbon source. Furthermore, we show that glycerol utilization uses a conserved 'classical' GK, which is homologous to GlpK, for glycerol phosphorylation, followed by G3P oxidation, which is catalysed by an unusual G3PDH enzyme, resembling a truncated version of GlpA in

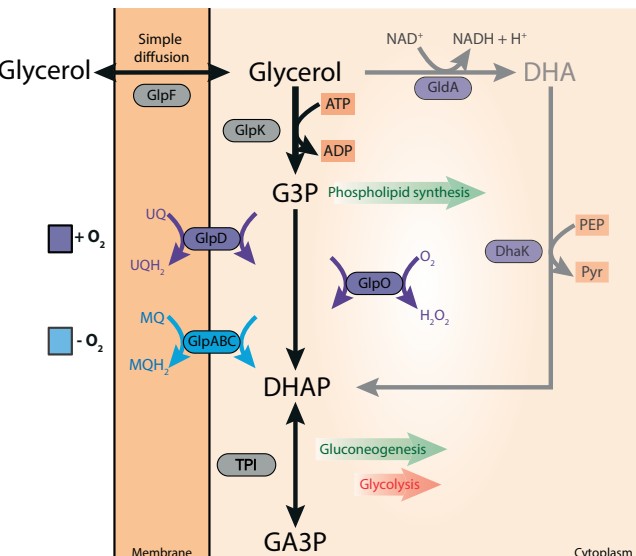

**Fig. 1 | Biochemical pathways involved in glycerol conversion in Bacteria and Eukarya forming dihydroxyacetone phosphate (DHAP) which is channelled into central metabolism.** Glycerol metabolism starts with its uptake either via facilitated diffusion mediated mainly by the glycerol uptake facilitator (GlpF) (encoded by the glpF gene) or via (protein-independent) simple diffusion through the cytoplasmic membrane. Following uptake, glycerol is converted finally to DHAP via two basic pathways: In respiring organisms, glycerol is first phosphorylated by the glycerol kinase (GlpK) (encoded by the glpK gene) to G3P which is further oxidized by two different membrane bound FAD-dependent G3P dehydrogenases (G3PDH), i.e. GlpD (encoded by the glpD gene, e.g. aerobic conditions E. coli, mitochondria) and GlpABC (encoded by the glpA, B, and C genes, e.g. anaerobic conditions E. coli, Haloarchaea). Electrons are transferred via the G3PDH bound FAD cofactor to ubiquinone (UQ) by GlpD, or to menaquinone (MQ) by GlpABC components in the respiratory chain. A third mechanism of G3P oxidation mainly known from aerotolerant lactic acid bacteria as well as Mycoplasma species is catalysed by a soluble, cytoplasmic FAD-dependent G3P oxidase (GlpO, encoded by the glpO gene) which directly utilizes molecular oxygen as electron acceptor. The second basic glycerol converting pathway is known from anaerobic, fermentatively growing organisms. Here, glycerol is first oxidized via an $NAD^+$-dependent G3PDH to dihydroxyacetone (GldA, encoded by the gldA gene) which is subsequently phosphorylated by dihydroxyacetone kinase (DhaK, encoded by dhaK gene) with phosphoenolpyruvate (less frequently ATP) as phosphoryl donor. In all cases, DHAP is further degraded via the lower common shunt of the ED and EMP pathway or utilized for gluconeogenesis. G3P serves as building block for phospholipid/membrane synthesis in Bacteria and Eukarya.

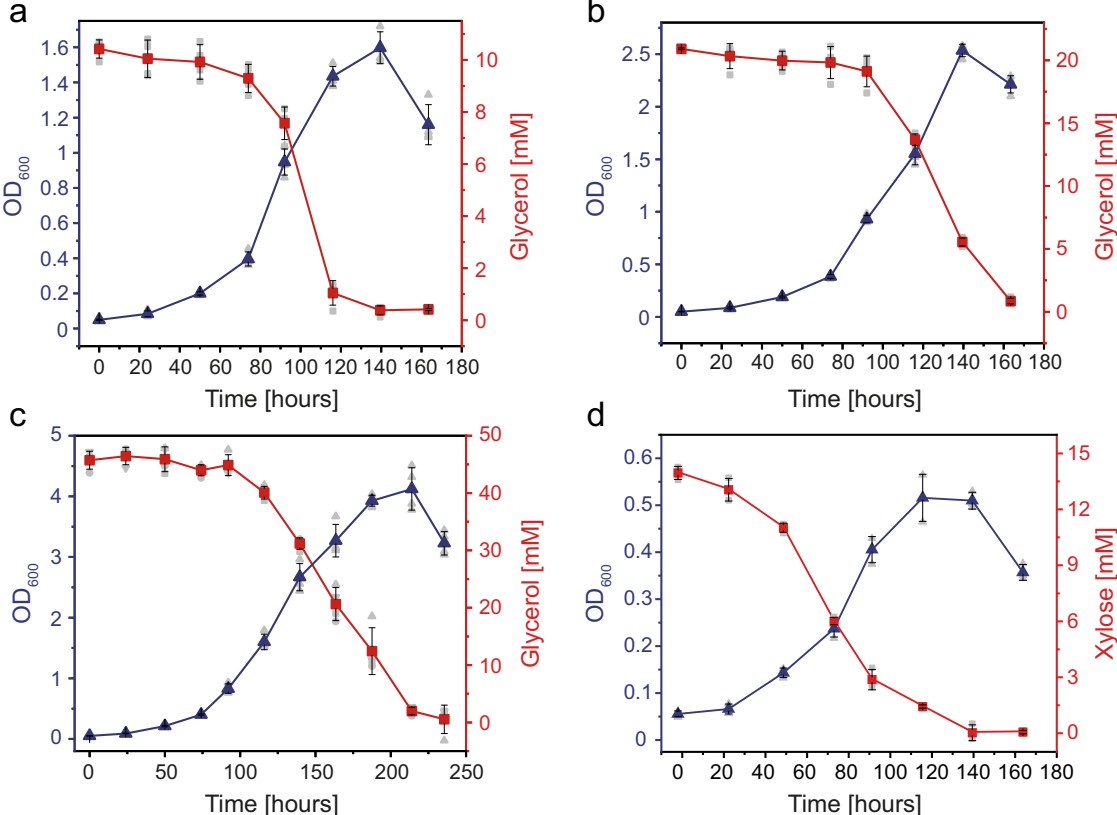

**Fig. 2 | Growth and substrate consumption of *S. acidocaldarius* MW00G.** Cells were grown on Brock's basal medium containing 10 mM (**a**), 20 mM (**b**), and 40 mM (**c**) glycerol as sole carbon and energy source under aerobic conditions. For comparison the growth with 14 mM D-xylose (**d**) was studied. Growth was monitored as increase in $OD_{600}$. Substrate concentrations in the cell free supernatant were determined enzymatically as described in the materials and methods part. Experiments were performed in triplicate (n = 3, biological replicates) and error bars indicate the standard deviation (SD) of the mean and individual data points are shown as grey dots.

which membrane association is facilitated by a small carbon monoxide dehydrogenase subunit G (CoxG)-like protein. Finally, we reveal a diverse repertoire of G3PDHs across Archaea, featuring various interacting proteins, electron transfer pathways, and potential modes of membrane anchoring. Our results expand our knowledge on the diversity of mechanisms used for glycerol metabolism in Archaea and are likely to support future biotechnological uses of *S. acidocaldarius*.

## Results

### *S. acidocaldarius* glycerol catabolism involves GK and G3PDH

Following a period of adaption, *S. acidocaldarius* MW001 exhibited exponential growth when supplied with glycerol (10 mM, 20 mM, and 40 mM) as sole carbon and energy source. Notably, similar growth rates of approximately $0.0287 \pm 0.0005$ $h^{-1}$ were observed across all glycerol concentration, and cell densities up to an $OD_{600}$ of 4 (at 40 mM glycerol) were achieved, coinciding with complete consumption of glycerol (Fig. 2a–c, Supplementary Table 1). The adapted strain was named *S. acidocaldarius* MW00G. For comparison, *S. acidocaldarius* MW00G grew on 0.2% (w/v) D-xylose, a commonly used carbon source, to a much lower final $OD_{600}$ of 0.8 and with a slower growth rate of $0.0195 \pm 0.0005$ $h^{-1}$ (Fig. 2d).

Having shown that *S. acidocaldarius* can grow faster and to higher biomass yields (corresponding to increased yield coefficients) (Supplementary Table 1) on glycerol than on D-xylose, we then compared the cellular responses to these carbon sources using transcriptomics (RNA-Seq) and proteomics (LC-MS-MS). On glycerol, a total of 39 transcripts/proteins were significantly upregulated while 14 transcripts/proteins were downregulated with at least a log2-fold change of 2 (Supplementary Tables 2 and 3). Downregulation was observed for the Weimberg pathway for pentose degradation (*saci_1938*, α-ketoglutarate semialdehyde dehydrogenase; *saci_1939*, 2-dehydro-3-deoxy-D-arabinonate dehydratase) and for the sugar binding subunit of the xylose/arabinose transporter (*saci_2122*). This coincided with previous findings demonstrating this pathway to be up-regulated by D-xylose[43]. Other central carbohydrate metabolic pathways (e.g. branched ED pathway, tricarboxylic acid (TCA) cycle, and EMP) were unaffected. Notably, we found that glycerol induced a significant upregulation of the gene cluster *saci_2031-2034* (Fig. 3a) and its encoded proteins, including a putative G3PDH (Saci_2032, GlpA-like) with a downstream encoded protein annotated as CoxG (Saci_2031), as well as a putative GK (Saci_2033) and a putative glycerol uptake facilitator (GUF; Saci_2034). Both, *saci_2032-2031 and saci_2033-2034* form operons and are divergently oriented (Fig. 3b). In accordance with the regulation pattern of *saci_2031-2034*, we also observed the induction of enzymatic activities in soluble crude extracts of cells grown in glycerol compared to cells grown on D-xylose: GK activity increased from $0.06 \pm 0.002$ U $mg^{-1}$ to $0.89 \pm 0.048$ U $mg^{-1}$, whereas G3PDH activity increased from non-measurable to $0.12 \pm 0.006$ U $mg^{-1}$ activity (with DCPIP as electron acceptor) (Fig. 3c). Neither $NAD^+$-dependent G3P oxidation nor DCPIP- or $NAD^+$-dependent glycerol oxidation could be observed, indicating that no alternative pathway for glycerol dissimilation is present. Notably, the DCPIP-dependent G3PDH activity was not only located in the soluble fraction (crude extract) but also in the resuspended membrane fraction with $0.13 \pm 0.018$ U $mg^{-1}$ (Fig. 3c), whereas GK activity was exclusively found in the soluble fraction. In addition to *saci_2031-34*, we identified a second gene cluster (*saci_1117-1119*) encoding isoenzymes for GK (*saci_1117*), G3PDH (*saci_1118*) and the CoxG-like protein (*saci_1119*). However, in contrast to *Saci_2031-2034*, a homologue encoding a GUF (or other transporter) is missing from this cluster (Fig. 3b). Furthermore, this second cluster *saci_1117-1119* was only slightly upregulated in response to glycerol

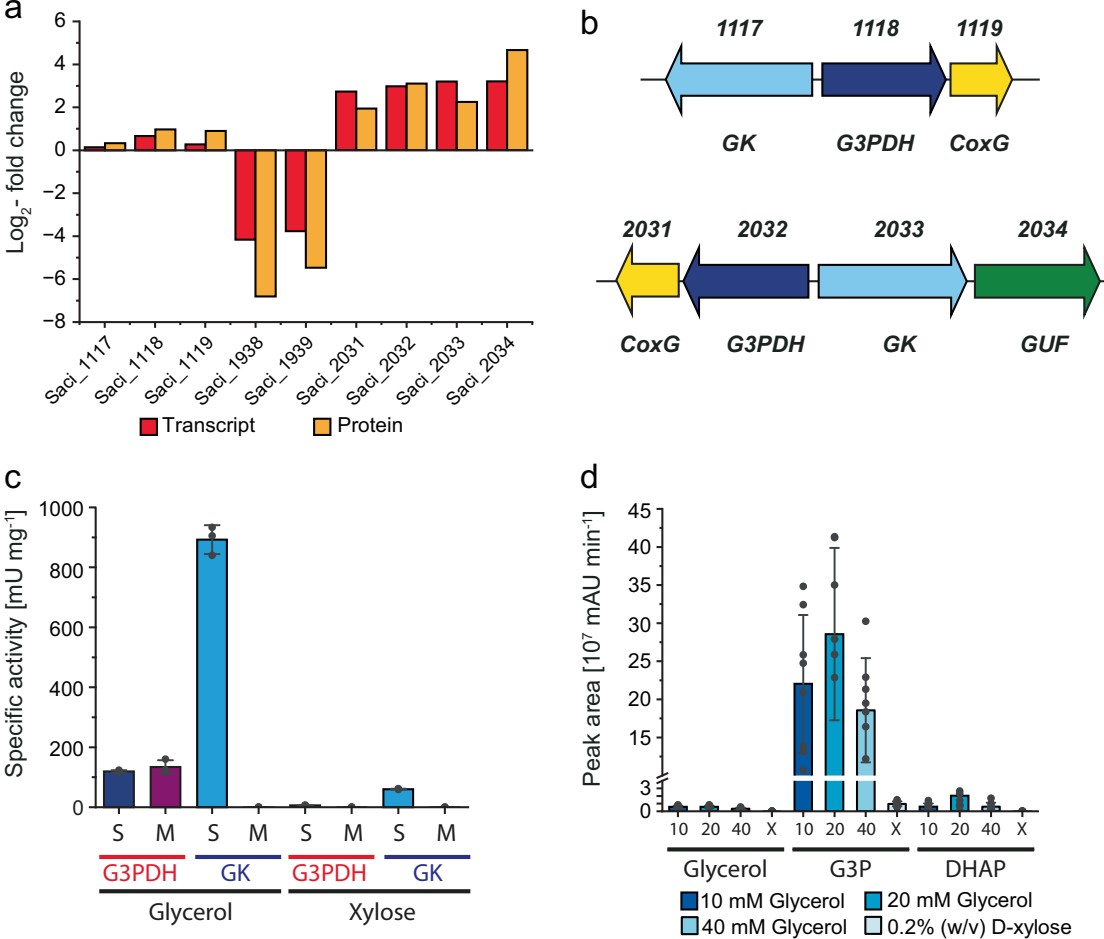

**Fig. 3 | Multi-omics and activity-based analyses of glycerol degradation in *S. acidocaldarius*. a** Log2-fold changes of transcript and protein levels of *saci_1117-1119, saci_1938-1939* and *saci_2031-2034* of *S. acidocaldarius* MW00G grown on 40 mM of glycerol in comparison to 0.2% (w/v) of D-xylose. **b** Genome organization. Divergent orientation of the two potential glycerol degradation gene clusters *saci_1117-1119* and *saci_2031-2034*. These clusters encode key enzymes including glycerol kinases (GK, GlpK), glycerol-3-phosphate dehydrogenases (G3PDH, GlpA), CoxG homologues, and, in the case of *saci_2031-2034*, a putative glycerol uptake facilitator (GUF). **c** GK and G3PDH activity in the soluble (S) and membrane fraction (M) (after cell lysis) of *S. acidocaldarius* MW00G grown on 40 mM glycerol in comparison to 0.2% (w/v) D-xylose. **d** Glycerol, G3P and DHAP determined by targeted metabolomics in *S. acidocaldarius* MW00G grown on glycerol (10 mM (10), 20 mM (20) and 40 mM (40) glycerol) compared to D-xylose (0.2% (w/v) (X)). All values represent the average of three (crude extract activities) or eight (metabolic analyses) independent measurements (biological replicates). Error bars represent the SD of the mean and individual data points are shown as dots.

compared to D-xylose, at both the transcript and protein level (Fig. 3a). We also used targeted LC(HILIC)-MS/MS-based metabolome analyses to further characterize glycerol metabolism in *S. acidocaldarius*. Focusing on central metabolic intermediates, especially those involved in glycerol catabolism, we found minimal levels of free glycerol in glycerol grown cells, while the concentrations of G3P and DHAP were substantially increased, compared to cells grown on D-xylose (Fig. 3d).

These multi-omics analyses demonstrate that *S. acidocaldarius* effectively utilizes glycerol as its sole carbon and energy source. Our data support a metabolic route for glycerol utilization in which following transport into the cell, likely involving the GUF (Saci_2034), glycerol is phosphorylated to G3P via GK (GlpK, Saci_2033), and further oxidized to DHAP by G3PDH (GlpA-like, Saci_2032). DHAP is then channelled either into the common lower shunt of the ED and EMP pathway for glycolysis or the upper EMP for gluconeogenesis.

**Purification and characterization of GK and G3PDH isoenzymes**
To further characterize the enzymes involved in glycerol metabolism in *S. acidocaldarius*, the two putative GKs Saci_1117 and Saci_2033 were homologously produced in *S. acidocaldarius* MW001 as C-terminally His- and Twin-Strep-tagged proteins, respectively, and purified. Saci_1117 and Saci_2033 comprise 498 and 497 amino acids with calculated molecular

masses of 55.6 and 55.3 kDa, respectively, coinciding well with the approximately 55 kDa determined by SDS-PAGE (Supplementary Fig. 1). The native molecular weight of ~110 kDa for both proteins determined by SEC indicated a homodimeric structure ($\alpha_2$) (Supplementary Fig. 2). Both isoenzymes showed GK activity in vitro, which followed Michaelis-Menten kinetics. For Saci_1117, we determined $V_{max}$ values of 45.7 ± 0.25 U mg$^{-1}$ and $K_M$ values of 0.024 ± 0.001 mM (glycerol) and 0.205 ± 0.024 mM (ATP) at 50 °C (Fig. 4, Table 1). The kinetic properties of Saci_2033, with a $V_{max}$ value of 88.2 ± 2.5 U mg$^{-1}$ and $K_M$ values of 0.024 ± 0.004 mM (glycerol) and 0.174 ± 0.019 mM (ATP), revealed a two-fold higher catalytic efficiency compared to Saci_1117 (Table 1). For additional information on substrate specificity, pH- and temperature optima, thermal stability, substrate and phosphoryl donor specificity and effect of fructose-1,6-bisphosphate see Supplementary Information 'In-depth comparison of structural, enzymatic, and regulatory properties of glycerol kinases across the three domains of life' (Supplementary Figs. 3 and 4, Supplementary Table 3).

We also characterized the in vitro activity of the two GlpA-like G3PDHs, Saci_1118 and Saci_2032. Heterologous expression in *E. coli* and purification of both N-terminally His-tagged G3PDHs, with calculated molecular mass of 46.9 kDa for Saci_1118 and 47.6 kDa for Saci_2032, yielded soluble, yellow proteins with subunit sizes of around 50 kDa (SDS-PAGE) and native molecular masses of 95 kDa (Supplementary Fig. 5), thus

**Fig. 4 | Kinetic characterization of the purified recombinant GK isoenzymes Saci_1117 and Saci_2033.** The kinetic properties of Saci_1117 (red circles) and Saci_2033 (blue squares) with ATP (**a**) and glycerol (**b**) were analysed using a continuous assay at 50 °C. The glycerol-dependent conversion of ATP to ADP was coupled to NADH oxidation via pyruvate kinase (PK) and lactate dehydrogenase (LDH), with the decrease in absorbance at 340 nm used to monitor the reaction. All experiments were performed in triplicate (n = 3, technical replicates), and error bars represent the SD of the mean and individual data points are shown as grey dots.

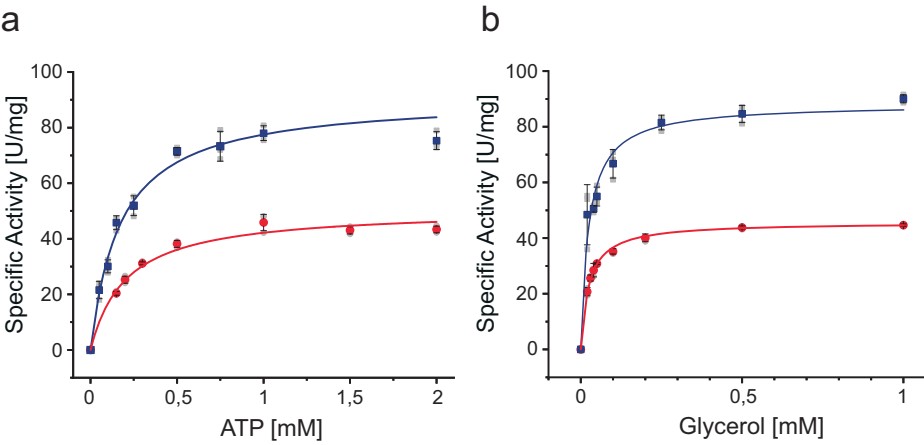

representing homodimers. The yellow colour indicated the presence of FAD cofactor, and a FAD content of two per homodimer was determined for both enzymes (Fig. 5a). After reduction with G3P, without adding an artificial electron acceptor, the FAD cofactor remained stable in its colourless, reduced state. To exclude G3P oxidase (GlpO) activity, the direct electron transfer from FAD to oxygen forming $H_2O_2$ was excluded using the 2,2'-azinobis-(3-ethylbenzothiazoline-6-sulfonate) (ABTS) assay ([25], data not shown). Saci_1118 displayed $K_M$ and $V_{max}$ values of 0.019 ± 0.003 mM (G3P) and 19.7 ± 0.73 U mg$^{-1}$, respectively, while Saci_2032 exhibited values of 0.055 ± 0.009 mM (G3P) and 44.5 ± 2.47 U mg$^{-1}$, with DCPIP serving as the artificial electron acceptor (Fig. 5b, Table 1). Also, ubiquinone-Q1 was confirmed as the electron acceptor, as evidenced by G3P-dependent decrease of absorption at 280 nm (Fig. 5c) and kinetic characterization (Fig. 5d, Table 1). Ubiquinone-Q1 is a water-soluble analogue of ubiquinones which are found in bacterial and mitochondrial respiratory chains. Its short side chain of one isoprenoid unit renders it water-soluble and thus suitable for aqueous biochemical assays, unlike the longer-chain ubiquinones (up to ten isoprenoid units comprising side chains) found in bacteria and eukaryotes, which are water-insoluble and unsuitable for such assays. For additional information on (co)substrate specificity, pH- and temperature optima, thermal stability, and effect of non-ionic detergents and phospholipids see Supplementary Information 'In-depth comparison of structural, enzymatic, and regulatory properties of glycerol-3-phosphate dehydrogenases across the three domains of life' (Supplementary Figs. 6 and 7, Supplementary Table 4).

Collectively, these analyses demonstrate that the GK and G3PDH enzymes of *S. acidocaldarius* are functional in vitro and reveal minor differences in catalytic activity between isoenzymes that may be relevant for glycerol metabolism under different conditions.

## CoxG homologues serve as membrane anchor for *S. acidocaldarius* G3PDHs

The ability of both *S. acidocaldarius* G3PDHs to reduce quinones suggests that these enzymes are associated with the cellular membrane. However, while the archaeal G3PDH enzymes share similarities with the GlpA subunit from the bacterial GlpABC complex, *S. acidocaldarius* does not encode a GlpC subunit, which mediates membrane anchoring in Bacteria. Notably, both genes encoding G3PDH (*saci_1118 and saci_2032*) form operons with a CoxG homologue (*saci_1119 and saci_2031*, respectively), suggesting a functional association. Furthermore, CoxG proteins have been shown to mediate membrane associations in Bacteria, although this function has never been demonstrated in archaeal cells or in connection to glycerol metabolism. However, both G3PDHs were active in vitro in the absence of their corresponding CoxG homologue, indicating that CoxG is not essential for G3PDH activity or electron transfer to quinones.

To elucidate a potential role of the CoxG homologues Saci_1119 and Saci_2031 as membrane anchors for G3PDHs, we heterologously co-overexpressed *saci_1118/saci_1119* and *saci_2032/saci_2031* from the pETDuet-1-vector in *E. coli*, which resulted in an enrichment of the respective G3PDH in the membrane fraction as monitored via immuno-detection using anti-His antibodies. By contrast, expression of *saci_1118* and *saci_2032* alone without the CoxG proteins resulted in G3PDH proteins being exclusively localized in the cytoplasmic fraction (Figs. 6a, b).

Upon homologous production of Saci_1119 and Saci_2031 CoxGs in *S. acidocaldarius* MW00G, both HA-tagged proteins also predominantly localized in the membrane fraction. Furthermore, they could be solubilized using DDM, suggesting that both CoxG homologues are indeed membrane-associated in *S. acidocaldarius* in vivo (Fig. 6c). Co-immunoprecipitation experiments utilizing magnetic beads coupled with Anti-HA antibodies further validated the interaction between C-terminally HA-tagged CoxGs Saci_1119 and Saci_2031 and their G3PDHs Saci_1118 and Saci_2032. MS analysis of interacting proteins unveiled a specific interaction of Saci_1119 with Saci_1118 and Saci_2031 with Saci_2032 (Supplementary Table 5). These findings indicate that the membrane interaction of G3PDHs is mediated by the CoxG homologues, which represents an additional function of CoxG proteins in Archaea and an uncommon mechanism of membrane anchoring of G3PDHs in *S. acidocaldarius*.

## Deletion of only one of two GK-encoding genes affects growth on glycerol

To elucidate the importance of individual GK paralogues in glycerol conversion, single deletion mutants Δ*saci_1117* and Δ*saci_2033*, as well as the double-deletion mutant Δ*saci_1117*Δ*saci_2033*, were constructed in the parental strain MW00G (Fig. 7). The Δ*saci_1117* mutant exhibited a reduced growth rate on glycerol compared to the MW00G strain (Fig. 7a), along with a slightly delayed glycerol consumption (Fig. 7b), and only a minor decrease in GK activity in crude extracts (Supplementary Fig. 8a). By contrast, growth of both the single Δ*saci_2033* and double Δ*saci_1117*Δ*saci_2033* mutant was completely abolished (Fig. 7a) and accordingly glycerol consumption was entirely blocked (Fig. 7b), highlighting the essential role of Saci_2033 for growth on glycerol. Since the single Δ*saci_2033* and double Δ*saci_1117*Δ*saci_2033* mutant did not exhibit any growth on glycerol, the effect on GK crude extract activity of the three mutants was determined in cells grown on N-Z-amine as alternative carbon source (Supplementary Fig. 8b). In Δ*saci_1117*, GK activity remained unaffected (115 ± 16 mU mg$^{-1}$), while only minor (20 ± 5 mU mg$^{-1}$) or no GK activity could be detected in Δ*saci_2033* and Δ*saci_1117*Δ*saci_2033*, respectively. This further confirms the significant contribution of Saci_2033 to the total GK activity.

Both single deletion strains, Δ*saci_1117* and Δ*saci_2033*, were complemented in trans by ectopic integration of the wildtype gene *saci_1117* and

**Table 1 | Kinetic characterization of the recombinant GK and G3PDH isoenzymes from *S. acidocaldarius***

| Enzyme | Substrate | $K_M$ [mM] | $V_{max}$ [U mg$^{-1}$] | $k_{cat}/K_M$ [mM$^{-1}$ s$^{-1}$] | Calculated mass [kDa] | Native mass [kDa] | Oligomeric state | Temperature optimum | pH optimum |
|---|---|---|---|---|---|---|---|---|---|
| Saci_1117[a] (GK) | Glycerol | 0.024 ± 0.001 | 45.7 ± 0.25 (170 ± 9)[b] | 6397 ± 507.4 | 55.6 | 105 | Homodimer | 75 °C[a] | 7 |
| | ATP | 0.205 ± 0.024 | 50.8 ± 2 | 229 ± 28.4 | | | | | |
| Saci_2033[a] (GK) | Glycerol | 0.024 ± 0.004 | 88.2 ± 2.5 (339.7 ± 7.3)[b] | 13129 ± 2205 | 55.3 | 110 | Homodimer | 75 °C[a] | 6.5 |
| | ATP | 0.174 ± 0.019 | 91 ± 4 | 485 ± 57.1 | | | | | |
| Saci_1118[c] (G3PDH) | G3P | 0.019 ± 0.003 | 19.65 ± 0.73 | 808 ± 131.4 | 46.9 | 95 | Homodimer | 70 °C | 6.5 |
| | Ubiquinone-Q1 | 0.086 ± 0.019 | 18.66 ±1.41 | 170 ± 39.7 | | | | | |
| Saci_2032[c] (G3PDH) | G3P | 0.055 ±0.009 | 44.5 ±2.47 | 642 ± 110.9 | 47.6 | 94 | Homodimer | 70 °C | 6.5 |
| | Ubiquinone-Q1 | 0.179 ± 0.075 | 21.1 ± 5.25 | 128 ± 58.9 | | | | | |

Parameter values are given ± standard deviation.
[a] Kinetic constants for the Saci_1117 and Saci_2033 GKs were determined at 50 °C.
[b] Specific activities were determined under $V_{max}$ conditions at 75 °C.
[c] Kinetic constants for the Saci_1118 and Saci_2032 G3PDH were determined at 70 °C.
For comparison to other characterized GKs and G3PDHs see Supplementary Tables 3 and 4.

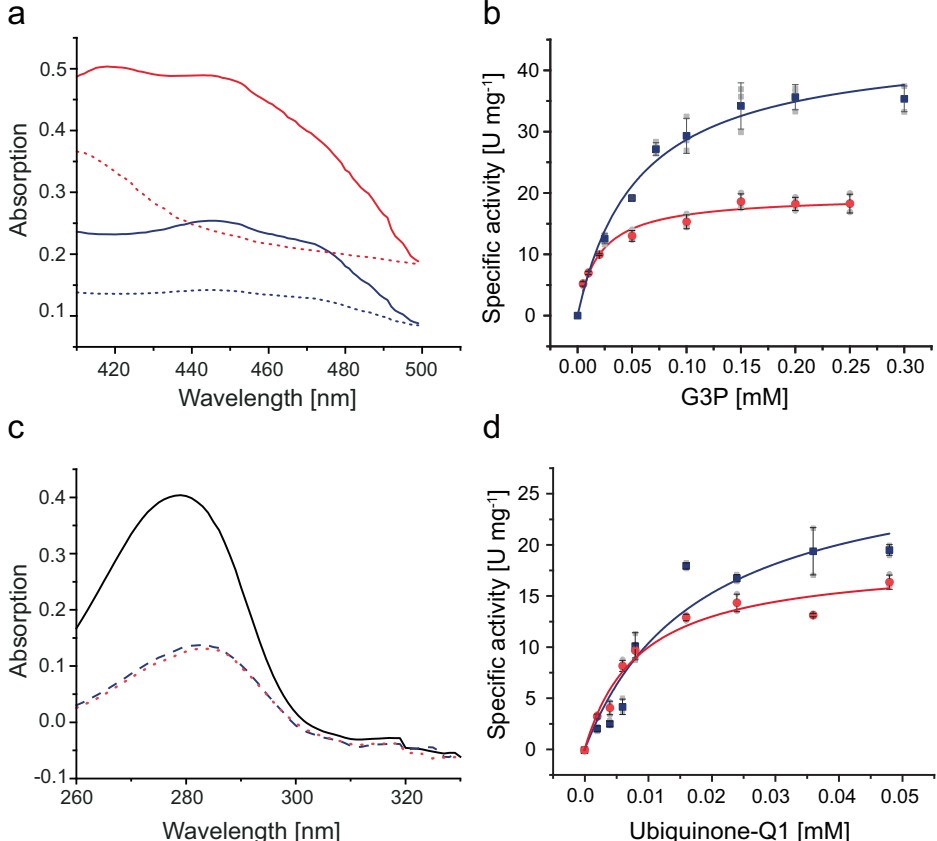

**Fig. 5 | Biochemical and kinetic characterization of the recombinant G3PDH isoenzymes Saci_1118 and Saci_2032. a** Absorption spectrum (400–500 nm) of Saci_2032 (blue solid line) and Saci_1118 (red solid line) at 70°C with bound FAD before addition of G3P. After addition of G3P (10 min), the loss of absorption at 450 nm indicates the G3P-dependent reduction of bound FAD in Saci_1118 (red dashed line) and Saci_2032 (blue dashed line). The maximal absorption at 450 nm after denaturation by SDS was used to calculate the amount of bound FAD per protein. **b** The kinetic properties of recombinant Saci_1118 (red circles) and Saci_2032 (blue squares) with G3P as substrate (0–0.3 mM) were determined in a continuous assay at 70 °C by coupling the oxidation of G3P to the reduction of the artificial redox active dye DCPIP following the decrease of absorbance at 600 nm. **c** Absorption spectra of ubiquinone-Q1 at 70 °C in the absence (solid line) and in the presence of Saci_1118 (red dotted line) or Saci_2032 (blue dashed line) and 200 μM of G3P after 60 seconds. Loss of absorption at 280 nm indicates reduction of ubiquinone-Q1 by G3PDH. **d** The kinetic properties of Saci_1118 (red circle) and Saci_2032 (blue square) with respect to ubiquinone-Q1 were determined at 70 °C in a continuous assay following the G3P-dependent reduction of ubiquinone-Q1 to ubiquinol-Q1 at 280 nm. Experiments were performed in triplicate (n = 3, technical replicates) and error bars indicate the SD of the mean and individual data points are shown as grey dots.

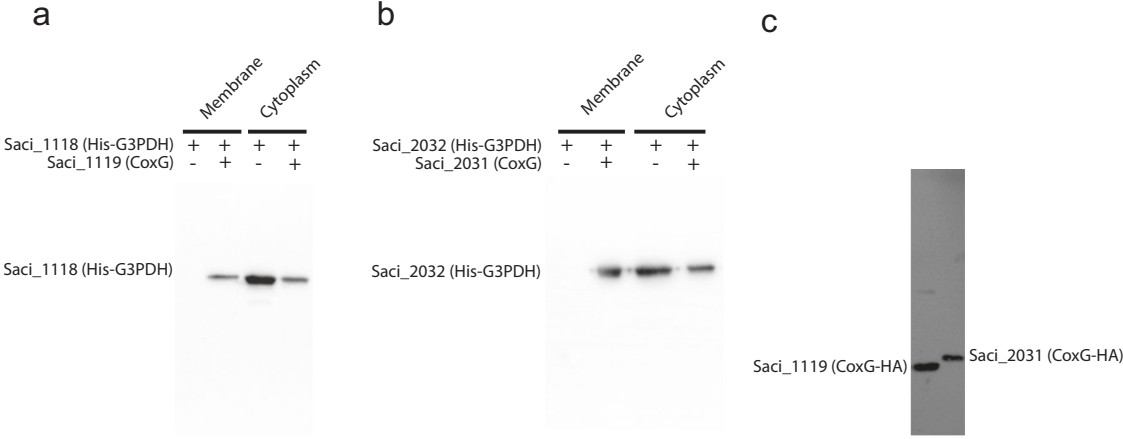

**Fig. 6 | CoxG homologues Saci_1119 and Saci_2031 facilitate membrane binding of their respective G3PDH isoenzymes Saci_1118 and Saci_2032 in *E. coli* and *S. acidocaldarius*.** The effect of heterologous co-expression of CoxGs on the localization of His-tagged G3PDHs in the cytoplasmic and membrane fractions of *E. coli* was analysed via western blotting, followed by immunodetection using an anti-His antibody. Only upon co-expression of CoxG (Saci_1119) and G3PDH (Saci_1118) (**a**) as well as CoxG (Saci_2031) and G3PDH (Saci_2032) (**b**) the respective G3PDH is found in the membrane fraction. **c** Isolated membrane fractions from *S. acidocaldarius* cells following homologous overexpression of HA-tagged CoxG homologues Saci_1119 (left lane) and Saci_2031 (right lane), utilizing the pSVAaraFX-HA vector, were analyzed. Western blotting and immunodetection with an anti-HA antibody clearly demonstrated the membrane localization of both CoxG homologues.

the *saci_2033/34* operon, respectively, into the *upsE* gene locus (*saci_1494*), under control of their respective native promoters. For both Δ*saci_1117* and Δ*saci_2033* mutants, complementation significantly restored growth (Fig. 7c), glycerol consumption (Fig. 7d) and crude extract activities to wild type levels (Supplementary Fig. 8a). Together, these results indicate that Saci_2033 is the primarily expressed and essential GK during growth on glycerol, whereas Saci_1117 plays a minor role under the conditions tested.

### Sequence and structural comparisons of G3PDHs

To further characterize the *S. acidocaldarius* G3PDH enzymes, we performed sequence analysis and structural comparison with other G3PDHs. As exemplified by Saci_2032 (Supplementary Figs. 9 and 10), these comparisons revealed homology between *S. acidocaldarius* and other archaeal G3PDHs in their N-terminal region of approximately 370 amino acids, with GlpDs, GlpAs and GlpOs, all exhibiting the D-amino acid oxidase (DAAO) superfamily fold (pfam01266), which includes for instance the glycine oxidase from *Bacillus subtilis* lacking any C-terminal extension (1ng3)[24,44]. These proteins consist of a 'glutathione-reductase-2' type FAD-binding domain and an antiparallel β-sheet-based substrate-binding domain[24]. Notably, both the FAD and the G3P binding site are conserved across these proteins (Supplementary Fig. 9). However, the comparison revealed substantial differences in the C-terminus of the various proteins. While glycine oxidase consists solely of the DAAO fold without any C-terminal extensions, the proteins GlpD, GlpO, GlpA as well as the archaeal homologues differ in length and domain organization of the C-terminus (Supplementary Figs. 10–12, for detailed discussion see Supplementary Information 'Extended analysis of sequence and structural comparisons of G3PDHs'). For example, the four highly conserved cysteine residues for FeS cluster binding, present in the C-terminus of GlpA sequences, are absent in GlpDs, GlpOs and the *S. acidocaldarius* G3PDHs (Supplementary Fig. 9). These structural differences in the C-terminal domains of the various proteins are likely correlated with their distinct functions. In GlpD, the C-terminus was shown to play a role in dimer formation, whereas in GlpOs the C-terminus is not involved in dimerization[15,20]. Though no crystal structure is available for GlpA, the presence of a FeS cluster in its C-terminus suggests its involvement in electron transfer to GlpB, supported by studies demonstrating that GlpAB forms a catalytically active unit. Our structural predictions of the Saci_G3PDH complex, coupled with the finding that Saci_2032 alone forms a soluble dimer (Supplementary Fig. 13), suggest that the C-terminus might also contribute to dimer formation. However, the predicted structure shows a different spatial orientation of the DAAO domains compared to GlpD, which combined with the interaction between Saci_G3PDH and CoxG, may facilitate optimal electron transfer to the quinone pool in the cytoplasmic membrane in *S. acidocaldarius*.

### Distribution, sequence similarities and phylogenetic affiliation of GK and G3PDH isoenzymes from *S. acidocaldarius*

The identification of unusual features in the G3PDH enzymes of *S. acidocaldarius*, including a distinct C-terminus and a previously undescribed association with CoxG, prompted us to examine the distribution and composition of GK and G3PDH enzymes across Archaea more generally. Notably, previous studies have analyzed the distribution of glycerol degrading genes/enzymes in Archaea, describing enzyme sets comprising GlpK/GlpA in several euryarchaeal orders including Thermococcales, Halobacteriales, Archaeoglobales and Thermoplasmatales, as well as in the crenarchaeal lineages such as Thermoproteales, Thermofilales, Desulfurococcales, and Sulfolobales[31,45]. However, GK/G3PDH homologues are only found in some representatives of these lineages, rather than being universally present or predominant across all Archaea, resulting in a patchy distribution. The overall sequence identity among archaeal GKs (40-70% identity) is higher compared to archaeal G3PDHs (20-50% identity). Furthermore, the closest homologues to *S. acidocaldarius* GKs were found in Thermococcales (61-68% identity), while the closest homologues to *S. acidocaldarius* G3PDHs were found in Thermoproteales and had lower sequence similarity (45-48% identity). The *S. acidocaldarius* GKs, show a high degree of sequence conservation not only to other archaeal homologues but also to bacterial ( > 50% sequence identity) and eukaryotic (~44%) enzymes, whereas archaeal G3PDHs are more diverse. These findings support that GKs and G3PDHs experienced distinct evolutionary histories and suggest that while GKs have a conserved function across organisms, G3PDHs have greater functional variability concerning interaction partners, membrane anchoring, and electron acceptors.

To better explore the differences between G3PDH enzymes, we used phylogenetic analyses of representative FAD-dependent G3PDH sequences across all domains of life, including Archaea, Bacteria and Eukarya. While our analyses yielded tree topologies consistent with previous reports[1,30,45], they also offer novel insights into the evolution and potential function of these enzymes (Fig. 8). For example, while we found that all G3PDHs cluster together within the DAAO superfamily, these enzymes could be subdivided based on their distinct C-terminal extensions: GlpAs and GlpDs form separate subgroups, with GlpOs exhibiting closer relation to GlpDs (falling

**Fig. 7 | Comparison of growth, glycerol consumption and GK activities in *S. acidocaldarius* parental (MW00G), GK deletion and complementation strains. a** Growth comparison of the parental strain MW00G with the single deletion strains *Δsaci_1117* and *Δsaci_2033*, and the double deletion strain *Δsaci_1117Δsaci_2033*. Cells were cultivated in Brock's basal medium containing 10 mM glycerol as the sole carbon and energy source. **b** Glycerol consumption of the different strains. **c** Growth and (**d**) glycerol consumption of the single deletion strains *Δsaci_1117* and *Δsaci_2033*, complemented in trans by ectopic integration of the wildtype genes *saci_1117* and the *saci_2033-34* operon, respectively, into the *upsE* gene locus (*saci_1494*) under control of their native promoters. For comparison, the growth of the single deletion strains and the parental strain as in (**a**) is included. GK activity of the parental MW00G, deletion mutants, and complementation strains was measured and is presented in Supplementary Fig. 8. The lower panel presents the colour code and symbols used throughout the figure. Experiments were conducted in triplicate (n = 3, biological replicates) and error bars indicate the SD of the mean and individual data points are shown as grey dots.

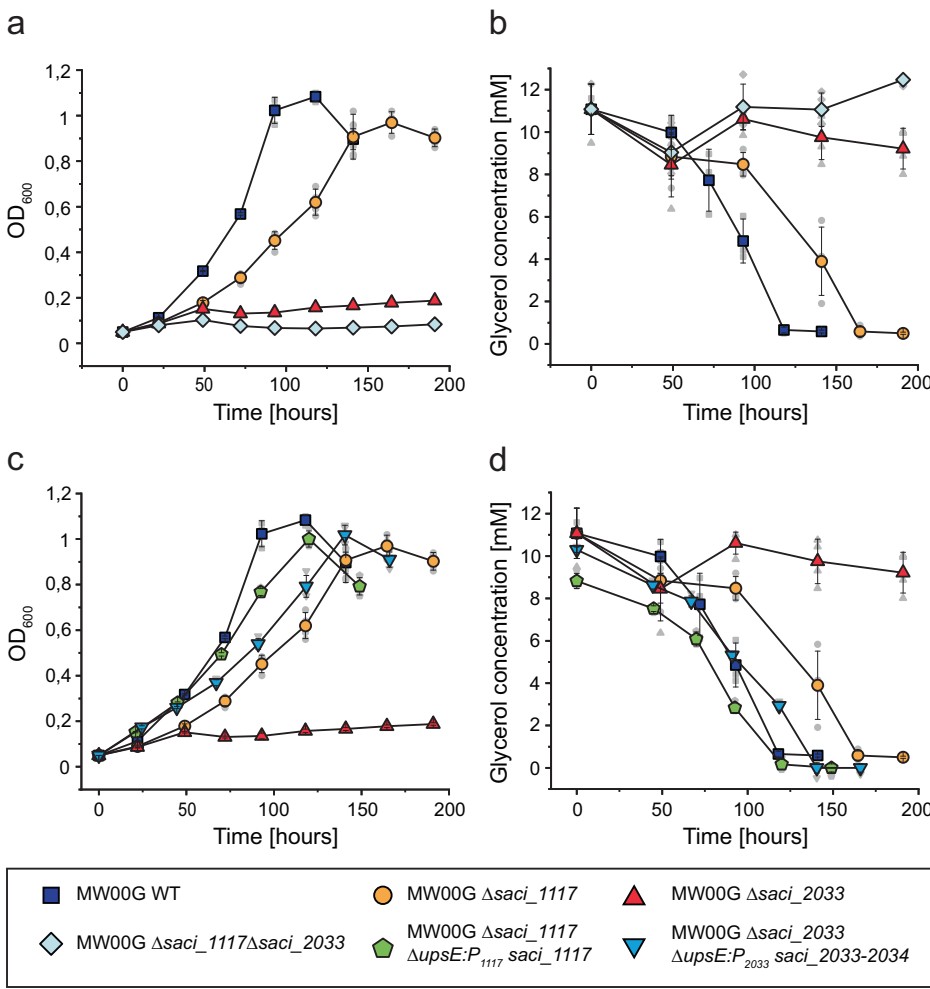

within the GlpD cluster) than to GlpAs. Furthermore, we found that G3PDHs from Sulfolobales, Thermoplasmatales, *Caldivirga maquilingensis* and *Vulcanisaeta moutnovskia*, as well as from Thermofilales form distinct subclusters within the GlpA cluster, distinct from canonical bacterial and haloarchaeal sequences. Notably, sequences from Thermococcales constitute a separate family within the DAAO superfamily, here designated as GlpTk, which also includes sequences from anaerobic Crenarchaea (Desulfurococcales), anaerobic Bacteria (e.g. *Thermotoga maritima* and *Clostridium perfringens*), and amitochondriate protists (*Giardia intestinalis*, *Entamoeba histolytica*, and *Spironucleus salmonicida*).

The observed phylogenetic clusters also coincide with conserved downstream gene synteny (Fig. 9): (i) Canonical *glpA* genes consistently co-occur with *glpB* and *C* genes; (ii) in the subcluster of *Thermofilum*-related species, *glpA* is followed by genes encoding putative counterparts of succinate dehydrogenase b, c, and d subunits, which are involved in membrane anchoring and electron transfer[46]; (iii) the *glpA* genes in the subcluster that includes *S. acidocaldarius* are associated with *coxG*; (iv) the genes in the *glpD* cluster lack obvious operon-like structures; (v) *Thermococcus*-related *glpTk* genes cluster with two genes encoding a NADH oxidase (NOX) and a molybdopterin oxidoreductase (MOX), and protein complex formation has been shown in vitro[35]. Notably, these three genes are fused in protists[47].

Collectively, the phylogenetic and gene synteny analyses suggest that G3PDHs form distinct complexes with a variety of proteins encoded downstream of the respective G3PDHs, which mediate membrane association and electron transfer. These observations suggest that the acquisition of glycerol utilization necessitated the concurrent evolution of tailored membrane anchoring and/or electron transfer systems to suit a variety of

metabolisms and terminal electron acceptors, which likely reflect distinct lifestyles across a range of organisms.

## Discussion

Glycerol is an abundant and important carbon source for many microbes and a highly regarded substrate for industrial applications. However, studies elucidating the pathways involved in glycerol degradation in Archaea have so far been mainly restricted to *H. volcanii*[1,28–30]. Here, we show that *S. acidocaldarius* grows on glycerol and demonstrate that glycerol involves a classical GK and an unusual G3PDH with a previously undescribed type of membrane anchoring via a CoxG-like protein.

*S. acidocaldarius* was previously reported not to grow on glycerol[37] and not to possess canonical G3PDH homologues[31]. However, it has been demonstrated that *S. acidocaldarius* can cleave short-chained triacylglycerols using esterases and can utilize short-chain fatty acids as carbon and energy sources for growth[42,48]. This metabolic capability leaves glycerol as a valuable byproduct, prompting us to investigate the capacity of *S. acidocaldarius* to utilize this substrate. We demonstrate that, after an adaptation period, *S. acidocaldarius* utilizes glycerol as sole carbon and energy source, even outperforming sugars such as D-xylose, as observed in *H. volcanii*[30,49]. Notably, a similar adaptation phase was needed to enable *Pseudomonas* spp. to grow on glycerol, which was attributed to transcriptional regulation induced by G3P[11]. In *S. acidocaldarius*, the higher growth yield on glycerol likely reflects its higher reduction state, suggesting a more efficient energetic coupling during glycerol utilization. This coincides with the up-regulation of the SoxEFGHIM gene cluster (*saci_2258-saci_2263*), one of three terminal oxidases with a higher $H^+/e^-$ ratio compared to the other two[50–53] (Supplementary Table 2).

**Fig. 8 | Phylogenetic affiliation of the *S. acidocaldarius* G3PDH isoenzymes Saci_2032 and Saci_1118 with selected GlpDs, GlpOs, and GlpAs from other Archaea, Bacteria and Eukarya.** GlpA homologues are shaded green, with the canonical bacterial and haloarchaeal GlpAs shown in dark green, the *Thermophilum* homologues in green and the *S. acidocaldarius* homologues in light green. The GlpD cluster shaded blue comprises the GlpDs and GlpOs highlighted in light and dark blue, respectively. The *Thermococcus*-like G3PDHs, designated as GlpTk are shown in orange. Organism names and uniprot accession numbers are given. The evolutionary history was inferred by using the Maximum Likelihood method and Le/Gascuel_2008 model[82]. The model was selected based on the lowest Bayesian information criterion value using the 'Find best protein model' option implemented in the MEGA11 package. The tree with the highest log likelihood (−18468.08) is shown. The percentage of trees in which the associated taxa clustered together is shown next to the branches. Initial tree(s) for the heuristic search were obtained automatically by applying Neighbor-Join and BioNJ algorithms to a matrix of pairwise distances estimated using the JTT model, and then selecting the topology with superior log likelihood value. A discrete Gamma distribution was used to model evolutionary rate differences among sites (5 categories (+$G$, parameter = 1.8751)). The rate variation model allowed for some sites to be evolutionarily invariable ([+$I$], 2.12% sites). The tree is drawn to scale, with branch lengths measured in the number of substitutions per site. This analysis involved 55 amino acid sequences. All positions containing gaps and missing data were eliminated (complete deletion option). There were a total of 262 positions in the final dataset. Evolutionary analyses were conducted in MEGA11[80].

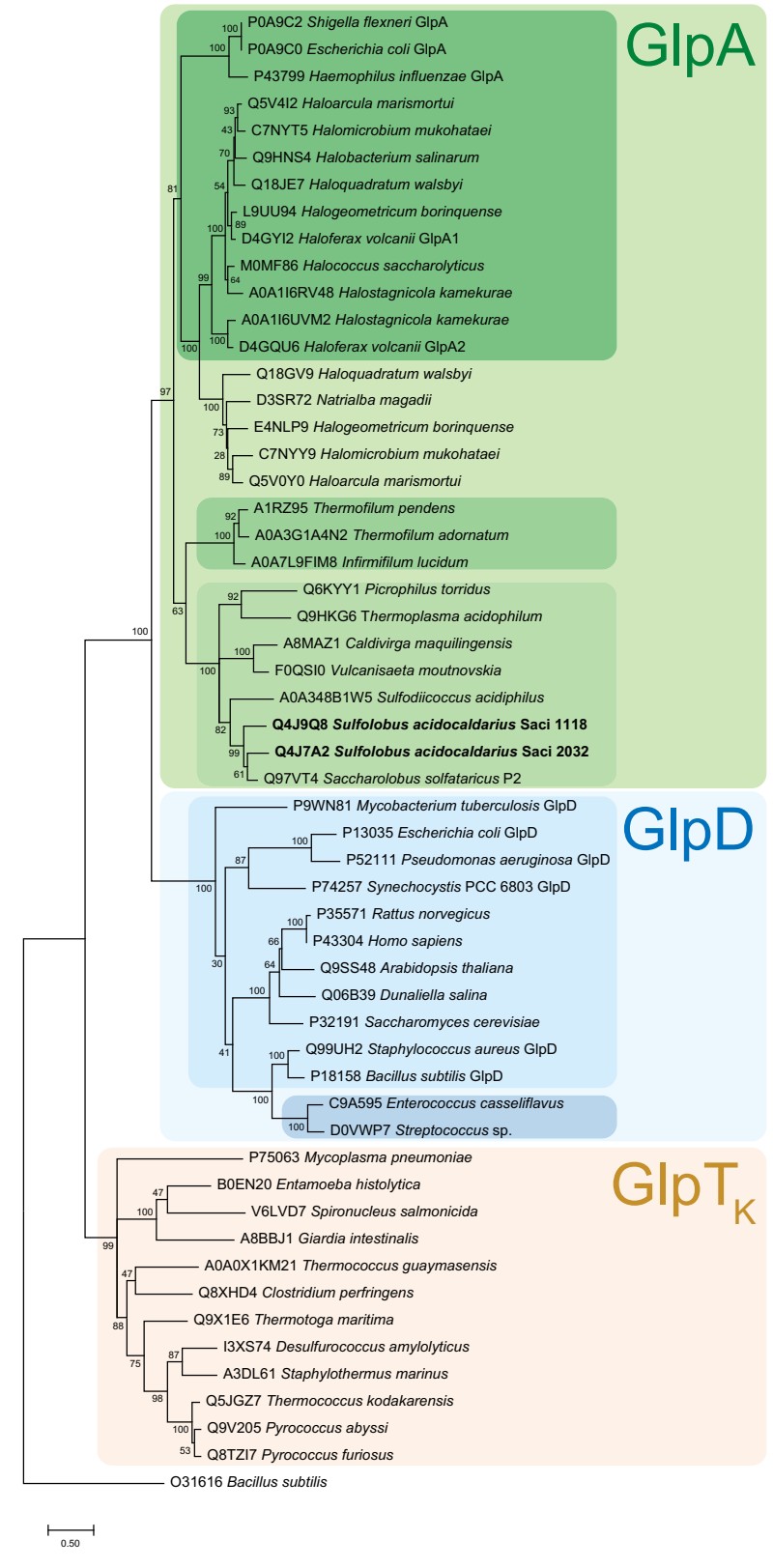

Our multi-omics analyses revealed that glycerol uptake in *S. acidocaldarius* involves a GUF homologous to GlpF, suggesting a mechanism similar to that observed in other organisms[3]. Additionally, simple diffusion may also contribute to glycerol uptake, as observed in Bacteria[4]. Notably, GlpF homologues are less abundant in Archaea than GKs and G3PDHs[1,30], suggesting that some organisms utilize alternative transporters, which is supported by the gene neighbourhoods of the GKs and G3PDHs often including genes encoding for putative MFS (major facilitator superfamily) transporters. In *S. acidocaldarius*, alongside the *glpF* gene, we observed that glycerol induced the up-regulation of a putative ABC transporter (*saci_1762-1765*), suggesting its potential involvement in glycerol transport.

**Fig. 9 | Comparison of different G3PDHs from Bacteria and Archaea.** The respective genome neighbourhood, the derived reconstructed protein complexes with potential interaction partners, membrane or cytoplasmic localization and cofactor content, as well as the catalytic oxidoreductase-monomer 3D-structure of the G3PDHs GlpABC (*E. coli* (**a**), GlpA-SdhBCD (*T. pendens*) (**b**), GlpA-CoxG (*S. acidocaldarius*) (**c**), GlpD (*E. coli*) (**d**) GlpO (*Streptocoocus*) (**e**) and GlpTk-NOX-MOX (*T. kodakarensis*) (**f**) are shown. For colour code see Fig. 8. For all 3D structures the principal FAD-binding G3P-oxidoreductase domain is coloured in grey while the highly divergent C-terminus is coloured according to Fig. 8.

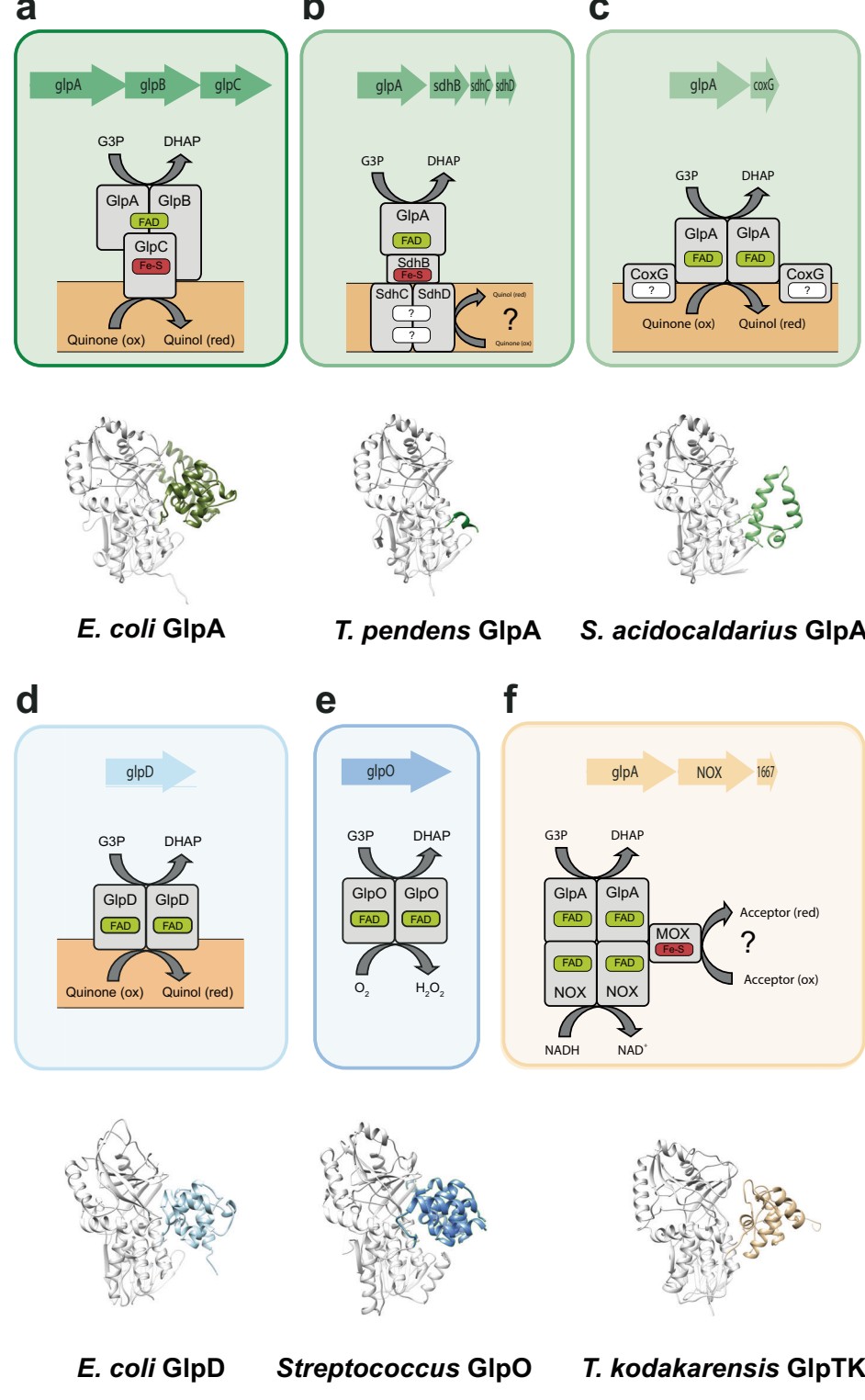

We found that in *S. acidocaldarius*, glycerol degradation follows the GK/G3PDH pathway, leading to the production of DHAP, which is then channelled into central metabolism (Fig. 10). While this pathway is similar to that used in *H. volcanii*[29], these organisms used distinct G3PDH enzymes for G3P oxidation. In *H. volcanii*, FAD-dependent G3P oxidation with electron transfer to the quinone occurs via a 'classical' bacterial-like 'anaerobic' GlpABC, whereas in *S. acidocaldarius* this process is catalysed by a structurally unusual G3PDH, in which membrane association is mediated by a CoxG-like subunit, which represents a previously undescribed function for CoxG proteins in Archaea and glycerol metabolism in general. Whereas in anaerobic bacteria and aerobic halophilic archaea, menaquinone derivatives serve as electron acceptors, in Sulfolobales, including *S. acidocaldarius*, the caldariellaquinone[54] is the primary quinone and thus likely functions as an electron acceptor for G3P oxidation in vivo (Fig. 10).

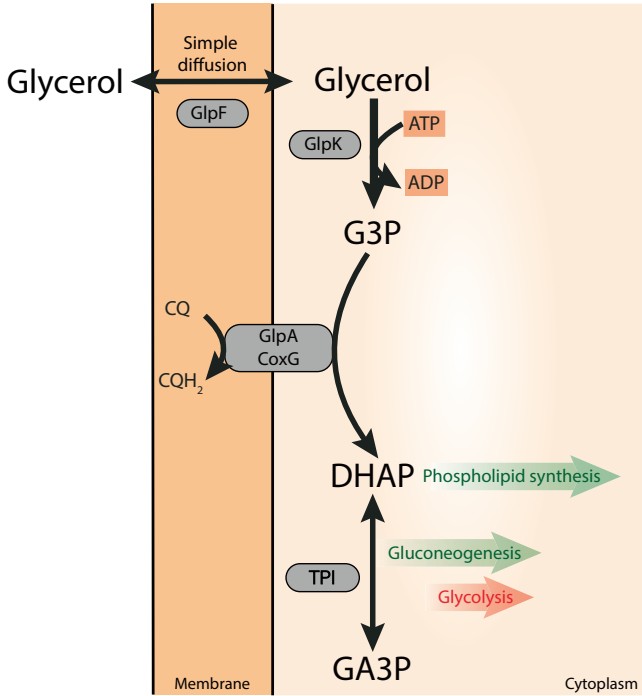

**Fig. 10 | Glycerol metabolism in *S. acidocaldarius*.** Glycerol uptake likely involves the GlpF-like glycerol uptake facilitator Saci_2034 and eventually simple diffusion through the cytoplasmic membrane. Additional uptake systems like the ABC transporter (Saci_1762-1765) might also contribute. Intracellularly, glycerol is phosphorylated by GK (Saci_2033), homologous to the bacterial GlpK, forming G3P which is then oxidized by a membrane bound, unusual GlpA-like FAD-dependent G3PDH (Saci_2032). Saci_2032 is anchored to the membrane by the CoxG homologue Saci_2031 forming a complex that transfers electrons to the caldar-iellaquinone (CQ) yielding dihydroxyacetone phosphate (DHAP). DHAP can be used for phospholipid synthesis (via G1P in Archaea), and - with concomitant triosephosphate isomerase (TPI, Saci_0117) mediated conversion to glyceraldehyde-3-phosphate (GA3P) - for gluconeogenesis (green arrows), or glycolysis (red arrow). The paralogues Saci_1117 (GK), Saci_1118 (G3PDH), and Saci_1119 (CoxG) might contribute to glycerol degradation under different growth conditions (for details see discussion section).

Our findings also show that *S. acidocaldarius* encodes two gene clusters for glycerol metabolism, *saci_1117-1119* and *saci_2031-2034*, which exhibit a similar organization (Fig. 3). However, these clusters differ in the presence of the GUF encoding gene *saci_2034*. Furthermore, only the *saci_2031-2034* gene cluster was substantially up-regulated on glycerol, and only the GK Saci_2033 was essential for growth on glycerol, indicating that the *saci_2031-2034* gene cluster is primarily involved in glycerol degradation under the tested conditions. The presence of two GK/G3PDH pairs is rare in both Archaea and Bacteria, although some species, including some Haloarchaea, encode two G3PDHs and a single GK[1,10,11,30]. In *H. volcanii*, only one of two G3PDH isoenzymes (GlpABC) was essential and up-regulated by glycerol[30,55]. High constitutive levels of G3PDH activity have been reported, with approximately a two-fold upregulation in the presence of glycerol. This is consistent with the finding that glycerol serves as the primary carbon source in halophilic environments[29,32]. In *E. coli*, the two G3PDHs, GlpD and GlpABC, are known to be primarily expressed under different environmental conditions[10,15,17]. Similarly, studies in species with two GK paralogues suggest functional specialization[1,55,56]. In *S. acidocaldarius*, the two G3PDH and GK paralogues exhibit very similar functional parameters and might thus be responsible for glycerol degradation under different growth conditions. In contrast to the *saci_2031-2033* GK/G3PDH/CoxG encoding genes co-occurring with the GlpF glycerol transporter, the genomic context of *saci_1117-1119* comprises genes which might be involved in lipid hydrolysis and fatty acid β oxidation[42]. On solid media

containing triglycerides like tributyrine, *S. acidocaldarius* was shown to be active and able to cleave these lipids by means of esterases encoded in the *saci_1103-1126* gene neighbourhood, i.e. Saci_1105 and Saci_1116[48]. Thus, this gene cluster might be involved in the glycerol utilization during lipid breakdown, but further analyses are needed to elucidate the specific functions of each glycerol gene cluster.

Our analyses suggest that *S. acidocaldarius* GKs have a homodimeric structure, which is consistent with some other GKs from Bacteria and Eukarya. Similarly, the *S. acidocaldarius* G3PDHs seem to form catalytically active homodimers, and directly reduce the ubiquinone Q1, resembling bacterial and eukaryotic GlpDs. Additionally, their FAD content of one per monomer aligns more closely with GlpDs[15,57,58]. GlpDs are monotopic membrane proteins that transfer electrons to the quinone pool, relying on membrane interaction, phospholipids, or non-denaturing detergents for conformational integrity and thus full activity[15]. By contrast, *S. acidocaldarius* G3PDHs alone are soluble and active without membrane association or detergent addition. We show that these G3PDHs interact with CoxG-like proteins, and that the G3PDHs only associate with the membrane when co-expressed with the CoxG-like proteins. The quinone reactivity of both G3PDH dimers in the absence of CoxG suggests that CoxG is not involved in electron transfer, but rather anchors the protein to the membrane. Our findings are consistent with the membrane-anchoring ability of CoxG homologues, including that from *Oligotropha carboxydovorans*, which has been shown to recruit the carbon monoxide dehydrogenase from the aerobic xanthin oxidase type to the membrane without enzymatic implications[59]. A similar function for a CoxG homologue in membrane anchoring of 3-hydroxypyridine dehydrogenase of *Ensifer adhaerens* HP1 has also been proposed[60].

The identification of unusual G3PDH isoenzymes in *S. acidocaldarius* led us to investigate the distribution, sequence similarities, and phylogenetic affiliation of GK and G3PDH enzymes across Archaea. Enzyme sets comprising GlpK/GlpA for glycerol degradation are widespread among various euryarchaeal orders, such as Thermococcales, Halobacteriales, Archaeoglobales, and Thermoplasmatales, as well as crenarchaeal lineages like Thermoproteales, Thermofilales, Desulfurococcales, and Sulfolobales. However, the evolutionary history of GKs and G3PDHs in Archaea appears divergent. In contrast to GK, which seems more widespread and conserved across archaeal lineages, the distribution of G3PDH homologues across these lineages appears patchy, suggesting a complex evolutionary history. While GK likely originated from horizontal gene transfer from Bacteria, G3PDH may have been present in the universal common ancestor, as well as in bacterial and archaeal common ancestors[45]. Our sequence analysis revealed high conservation among *S. acidocaldarius* GKs, belonging to the FGGY family within the sugar kinase/HSP70/actin superfamily, with significant similarity to both archaeal and bacterial enzymes, as well as eukaryotic counterparts. Our structural and sequence analyses of G3PDHs also highlighted strong homology in the N-terminal region among *S. acidocaldarius* and other archaeal G3PDHs (GlpAs) with canonical bacterial and haloarchaeal GlpAs (GlpABC complex), GlpDs, and GlpOs, all exhibiting the D-amino acid oxidase (DAAO) superfamily fold. However, despite conservation in the FAD and G3P binding sites, significant variability exists in the C-terminal domains of archaeal G3PDHs, likely correlating with their distinct functions. For example, the C-terminus of the canonical GlpA contains a bfd-like domain with an FeS cluster, suggesting involvement in electron transfer to GlpB, while GlpD and GlpO lack such extensions, consistent with their roles as homodimers.

Our additional phylogenetic and gene synteny analyses provide additional support for these structural differences among different G3PDH subgroups. However, the extent of diversity observed in Archaea is unexpected, especially when contrasted with the pattern observed in Bacteria. In Bacteria, canonical *glpA* genes consistently co-occur with *glpB* and *glpC*, mirroring the pattern found in Haloarchaea. By contrast, among various other archaeal lineages, distinct patterns of gene co-occurrence emerge. For instance, in *Thermofilum*-related species, *glpA* genes are associated with genes encoding succinate dehydrogenase subunits. In *Thermococcus*-related

species, *glpA* genes cluster with genes encoding NADH oxidase and molybdopterin oxidoreductase. Lastly, in the *S. acidocaldarius* subgroup, *glpA* genes are consistently linked with *coxG*. These findings suggest that the acquisition of glycerol utilization required the simultaneous evolution of specialized membrane anchoring and/or electron transfer systems to align with the organism's lifestyle, metabolism, and terminal electron acceptors.

Our findings will also likely have important implications for future biotechnological applications. The utilization of extremophiles such as *S. acidocaldarius* offers several benefits for industrial biotechnology, including increased reaction rates and improved substrate solubility that promote higher biomass conversion, and a reduction of microbial contamination at higher temperatures and low pH, making antibiotics obsolete[38]. *S. acidocaldarius* is well established as an industrial chassis due to the availability of its genome sequence and an extensive genetic toolbox, detailed metabolic characterization, and existing bioprocesses. Importantly, glycerol is generated during biodiesel production, and the rapid growth of biofuels has led to a significant oversupply of glycerol and declining prices, rendering it a highly regarded wasteful by-product that can support the production of a range of value-added products. Therefore, our demonstration that *S. acidocaldarius* grows efficiently on glycerol, coupled with our detailed characterization of the metabolic pathways involved in glycerol utilization, are likely to facilitate the development of engineered strains that can support the production of a range of valuable products from this cheap source material.

In summary, we show that glycerol degradation in *S. acidocaldarius* proceeds via the GK/G3PDH pathway involving 'classical' bacterial-like GKs and G3PDHs homologous to GlpA that are remarkably different from their bacterial counterparts. The G3PDHs from *S. acidocaldarius* lack the B and C subunits present in bacterial GlpABC complexes, differ in the C-terminal domain of the catalytic subunit, and are anchored to the membrane via CoxG homologues. Of two paralogous GK/G3PDH copies in the *S. acidocaldarius* genome, only *saci_2031-2033* is highly upregulated and essential during glycerol degradation. Furthermore, sequence analyses suggest that glycerol metabolism in Archaea is much more versatile with respect to the G3PDHs, their interaction partners, modes of membrane association, and electron transfer mechanisms, underscoring the intricate adaptation of glycerol metabolism to diverse environmental niches and metabolic pathways.

## Materials and methods
### Cultivation of *S. acidocaldarius* strains
*S. acidocaldarius* MW001 (uracil auxotroph mutant)[61] was adapted to glycerol utilization. The adapted strain designated *S. acidocaldarius* MW00G was routinely grown at 76 °C under constant shaking at 120 rpm (New Brunswick Innova 44127 incubator, Eppendorf, Hamburg, Germany) in Brock's basal salt medium[62], supplemented with 10 µg mL$^{-1}$ of uracil and 10 mM, 20 mM, or 40 mM glycerol, or for comparison with 0.2% (w/v) D-xylose as sole carbon and energy source. Growth was monitored as increase in $OD_{600}$ over time. *S. acidocaldarius* MW00G deletion mutant strains were pre-grown in Brock medium with 0.1% (w/v) N-Z-amine and 0.2% (w/v) dextrin and then subcultured in Brock's basal salt medium containing 10 µg mL$^{-1}$ of uracil and 10 mM glycerol. The glycerol concentration in the medium was determined enzymatically using the recombinant glycerol kinase (Saci_2033, see below), pyruvate kinase (PK) and lactate dehydrogenase (LDH) (both from rabbit muscle; Merck, Darmstadt, Germany). For each mole of glycerol, one mole of NADH is oxidized, with the reaction monitored at 340 nm in 96-well plates (BRANDplates®, BRAND, Wertheim, Germany). A NADH calibration curve (0–0.7 mM NADH) was used for quantification, and measurements were conducted using a Tecan Infinite M200 plate reader (Tecan Group AG, Männedorf, Switzerland). The reaction was performed in 200 µL total volume at 42 °C in 0.1 M MOPS-KOH buffer pH 6.5, 0.6 mM NADH, 1 mM PEP, 5 mM ATP, 5 mM $MgCl_2$, 5.5 U LDH, 2.8 U PK, 0.4 U Saci_2033, using 20 µL (diluted) samples with a glycerol concentration of at most 0.5 mM. For multi-omics analyses, *S. acidocaldarius* MW00G was pre-grown on minimal medium with the indicated carbon source (10 mM,

20 mM, 40 mM glycerol, 0.2% (w/v) D-xylose) or 0.1% (w/v) N-Z-amine (control) as described above. With these precultures, 400 mL main cultures (four replicates) were inoculated to an initial $OD_{600}$ of 0.05 and grown to exponential phase ($OD_{600}$ of 0.8). Culture samples were cooled down with liquid nitrogen and cells were harvested by centrifugation at $9000 \times g$, 15 min, 4 °C and stored at −70 °C for transcriptomic, proteomic and metabolomic analyses.

### RNA-seq
*RNA isolation, library preparation, and next-generation cDNA sequencing -* RNA was isolated using Zymo Direct-zol RNA Miniprep kit following the manufactures instructions. The RNA quality was checked by Trinean Xpose (Gentbrugge, Belgium) and the Agilent RNA Nano 6000 kit using an Agilent 2100 Bioanalyzer (Agilent Technologies, Böblingen, Germany). Pan-Archaea riboPOOL kit from siTOOLs Biotech was used to remove the rRNA. TruSeq Stranded mRNA Library Prep Kit from Illumina was applied to prepare the cDNA libraries. The cDNAs were sequenced paired end on an Illumina NextSeq 500 system (San Diego, CA, USA) using 74 bp read length mid output.

*Bioinformatics data analysis, read mapping and analysis of differential gene expression*—The paired-end cDNA reads were mapped to the *S. acidocaldarius* DSM 639/MW001 genome sequence (accession number CP000077.1) using bowtie2 v2.2.7. with default settings for paired-end read mapping. All mapped sequence data were converted from SAM to BAM format with SAMtools v1.3[63] and imported to the software ReadXplorer v.2.2[64].

Differential gene expression analysis of four replicates including normalization was performed using Bioconductor package DESeq2[65] included in the ReadXplorer v2.2 software[64]. The signal intensity value (A-value) was calculated by the log2 mean of normalized read counts and the signal intensity ratio (M-value) by the log2 fold change. The evaluation of the differential RNA-seq data was performed using an adjusted P-value cutoff of P ≤ 0.01 and a signal intensity ratio (M-value) cutoff of ≥2 or ≤−2. Genes with an M-value/log2 fold change outside this range and adjusted P ≤ 0.01 were considered as differentially up- or downregulated.

### Proteome
Samples for proteomic analysis were prepared using the single-pot, solid-phase-enhanced sample-preparation (SP3) strategy[66]. All buffers and solutions were prepared with mass spectrometry (MS)-grade water (Avantor, Radnor, PA, USA). Cell pellets were taken up in 200 µL 1× sample buffer (50 mM HEPES pH 8.0, 1% (w/v) SDS, 1% (v/v) NP-40, 10 mM TCEP, 40 mM chloroacetamide) and the samples were heated for 5 min at 95 °C prior to sonication with a Bioruptor UCD-200 (Diagenode, Seraing, Belgium) device for ten cycles of 1 min pulse and 30 sec pause at high power. The protein extracts were centrifuged (20,000 × g, room temperature (RT), 20 min) and the protein concentration of the cleared lysates was determined using the Pierce 660 nm Protein Assay Reagent (#22660; Thermo Scientific, Waltham, MA, USA) with the Ionic Detergent Compatibility Reagent (#22663; Thermo Scientific, Waltham, MA, USA) according to the manufacturers' instructions. Next, 15 µg of total protein in a volume of 47 µL 1× sample buffer was treated with 7 U of benzonase (#70746; Merck Millipore, Burlington, MA, USA) in dilution buffer (20 mM HEPES pH 8.0, 2 mM $MgCl_2$; 37 °C, 30 min, 1500 rpm) followed by the addition of iodoacetamide to a final concentration of 10 mM to complete alkylation of cysteine residues (RT, 30 min, 1500 rpm in the dark), resulting a sample volume of 50 µL. Then, 3 µL of a 50 µg µL$^{-1}$ 1:1 mixture of hydrophilic (#45152105050250) and hydrophobic (#65152105050250) carboxylate modified Sera-Mag™ SpeedBeads (Cytiva, Marlborough, MA, USA) that were washed twice with MS-grade water were added to the samples. Protein binding was induced by the addition of an equal sample volume of pure ethanol (24 °C, 20 min, 1500 rpm), the beads were collected using a magnetic stand. Beads were allowed to bind for at least 5 min before the supernatant was removed. The beads were washed thrice with 180 µL 80% (v/v) ethanol prior to the addition of the digestion enzyme mix (0.6 µg of trypsin (V5111; Promega,

Madison, WI, USA) and 0.6 µg LysC (125-05061; FUJIFILM Wako Pure Chemical, Osaka, Japan) in 25 mM ammonium bicarbonate) and incubation of the samples at 37 °C (16 h, 1500 rpm). Next day, the samples were briefly centrifuged (10 s, 2000 × $g$, RT) and placed on a magnet for 5 min. The clear solution containing the tryptic peptides was transferred to a new Eppendorf tube. The beads were taken up in 47 µL 25 mM ammonium bicarbonate (RT, 10 min, 1500 rpm). After incubation, the tubes were placed once more on a magnetic stand and after 5 min the clear supernatant was again collected and combined with the recovered peptide mix, followed by the addition of trifluoroacetic acid (TFA; 2% (w/v) final concentration) to the samples. Prior to LC-MS/MS analysis, peptides were desalted as described in ref. [67] with the only modification of not re-applying the flow-through to the $C_{18}$StageTips again.

## LC-MS/MS analysis

LC-MS/MS analysis of peptide samples was performed on an Orbitrap Fusion Lumos mass spectrometer (Thermo Scientific, Waltham, MA, USA) coupled to an EASY-nLC 1200 liquid chromatography (LC) system (Thermo Scientific, Waltham, MA, USA) that was operated in the one-column mode. The samples were separated on a self-packed analytical column (see supplementary File 1) filled with Reprosil-Pur 120 C18-AQ 1.9 µm (Dr. Maisch, Ammerbuch-Entringen, Germany) that was encased by a PRSO-V2 column oven (Sonation, Biberach an der Riß, Germany) and attached to a nanospray flex ion source (Thermo Scientific, Waltham, MA, USA). During data acquisition, the column oven temperature was adjusted to 50 °C. The LC was equipped with solvent A (0.1% (w/v) formic acid (FA), in water) and solvent B (0.1% FA (w/v), in 80% acetonitrile (ACN)) prepared from UHPLC (ultra-high-performance liquid chromatography)-grade solvents (Honeywell, Charlotte, NC, USA) as mobile phases. Peptides were directly loaded onto the analytical column with a maximum flow rate so that the set pressure limit of 980 bar would be exceed (usually around 0.5–0.8 µl min$^{-1}$) and separated by running gradients with different length and composition (for details see supplementary File 1).

The mass spectrometer was operated in the positive ion mode using Xcalibur software (v4.3.7.3.11). Precursor ions (MS$^1$) were scanned in the Orbitrap analyzer (FTMS; Fourier Transform Mass Spectrometry) with the internal lock mass option switched on (lock mass was 445.120025 m/z, polysiloxane[68]). Data-dependent product ion spectra (MS$^2$) were recorded in the ion trap. All relevant MS settings (resolution, scan range, AGC, ion acquisition time, charge states, isolation window, fragmentation type and details, cycle time, number of scans performed, and various other settings) can be found in supplementary File 1.

## Peptide and protein identification using MaxQuant and perseus

Recorded RAW data was analyzed with MaxQuant (v. 1.6.17.0 or v.2.0.1.0) using the default settings[69] with the Label-free quantification (LFQ)[70] and match between runs options activated. MS/MS spectra were searched against the UniProt *S. acidocaldarius* (DSM639) (UP000001018_330779.fasta; 2222 entries; downloaded on 2020-08-02) database. A search against a contaminants database as implemented in MaxQuant (contains known MS contaminants; 246 sequences) was included to estimate the level of contamination. For further data analysis and filtering of the MaxQuant output, LFQ intensities were loaded from the proteinGroups.txt file into Perseus (v. 1.6.14.0 or v.1.6.15.0)[71]. Contaminants as well as hits only identified based on peptides with a modification site and hits from the reverse database were removed. To allow comparison of the different treatment groups, biological replicates were combined into categorical groups and the data was transformed to the $\log_2(x)$ scale. For the full proteome analysis, only protein groups (PGs) with a valid LFQ intensity in at least three out of four replicates in a minimum of one categorical group were kept for further analysis.

For the identification of interaction partners of the CoxG homologues Saci_2031 and Saci_1119, the data was separately filtered to only keep hits with a valid LFQ intensity in at least two out of three replicates for samples containing the respective HA-tagged protein. The $\log_2$-fold change in normalized protein group quantities between the different categorical

groups was determined based on the mean LFQ intensities of replicate samples (relative quantification). To enable quantification, missing LFQ intensities were imputed from a normal distribution (full proteome analysis: width 0.3, down shift 1.8; identification of interaction partners: width 0.3, down shift 2.0). The statistical significance of the difference in LFQ intensity was determined via a two-sided Student's t-test. Full MS data for the comparative full proteome analysis and the identification of interaction partners of Saci_2031 and Saci_1119 can be found in Supplementary Files 2 and 3. Genes that were up- or downregulated with a log2-fold change ≥2 in response to growth on glycerol and are shared between the transcriptomics and proteomics analyses are reported in Supplementary Table 2. Proteins enriched by co-IP with a log2-fold change ≥2, excluding ribosomal proteins, are reported in Supplementary Table 5.

## Metabolome analyses

**Metabolite extraction.** The metabolite extraction was based on[72] with slight modifications. Cells were disrupted by resuspension in 500 µL of prechilled (−80 °C) methanol and the addition of 20 µL of internal standards (fructose 6-phosphate, arginine $^{13}C6$, and succinic acid d4), followed by vortexing and sonication both for 2 min. Samples were frozen at −80 °C for 5 min and the vortex and sonication steps were repeated. The methanol was evaporated by vacuum concentration (Concentrator plus, Eppendorf, Hamburg, Germany). When the extract was completely dry, 250 µL of water were added and the process was repeated. During the whole process the cells were kept on ice. After drying the water, extracts were resuspended in 100 µL ACN/water (85:15, v/v), sonicated 2 min, and vortexed 2 min. The mixture was centrifugated at 12,000 × $g$ for 2 min (MiniSpin centrifuge, Eppendorf, Hamburg, Germany). The supernatant was transferred to a LC vial. A quality control (QC) sample was prepared by pooling a same-volume aliquot from each sample to evaluate the performance and quality of the analytical instrumentation during the batch analysis. The samples were analyzed in a randomized order.

**LC-ESI-QTOF method for the targeted metabolomics analysis.** The samples were analyzed by a 1290 Infinity II LC instrument coupled to a 6546 LC/Q-TOF high-resolution mass spectrometer, and the ionization was performed using a Dual Jet Stream source in negative mode (Agilent Technologies, Waldbronn, Germany). For the LC separation, a iHILIC-(P)Classic (150 × 2.1 mm, 5 µm) (Hilicon, Umeå, Sweden) was used. The mobile phases consisted of 5 mM ammonium acetate, pH 5 (A) and ACN/5 mM ammonium acetate, 85:15 (v/v), pH 5 (B). The gradient elution was stablished as follows: 0 min, 90% B; 22 min, 80% B, 30 min, 65% B; 35 min, 65% B. The flow rate was set at 0.2 mL min$^{-1}$ and the column temperature was kept at 40 °C. The electrospray ion source parameters were: gas temperature, 200 °C; dry gas, 10 mL min$^{-1}$; nebulizer, 40 psi; sheath gas temperature, 300 °C; sheath gas flow, 12 L min$^{-1}$; fragmentor, 125 V; skimmer, 65 V; capillary voltage, 3000 V. Full scan-data-dependent acquisition was used to perform the tandem MS experiments. The retention time, the MS and the MS/MS spectra of the targeted metabolites were compared with commercial standards. The acquired data was processed by MS-Dial and Skyline software.

## Crude extract enzyme measurements

For the preparation of crude extracts, 50 mL cultures of *S. acidocaldarius* MW00G were grown using 40 mM glycerol or 0.2% (w/v) D-xylose as the sole carbon source until reaching an OD$_{600}$ of 0.6. The *S. acidocaldarius* MW00G deletion mutants were cultured in 50 mL media containing either 10 mM glycerol or 0.2% (w/v) N-Z-amine as the carbon source and grown to an OD$_{600}$ between 0.8 and 1. Cells were collected by centrifugation (4300 × $g$, 15 min, 4 °C), resuspended in 5 mL of 50 mM MES-KOH pH 6.5, 20 mM KCl, lysed by sonication (3 × 8 min, 60%, 0.5 s$^{-1}$, on ice) (UP200S, Hielscher Ultrasonics, Brandenburg, Germany), and cell debris was removed by centrifugation (4300 × $g$, 30 min, 4 °C). The supernatant was used as crude extract for enzyme activity measurements of GK. To separate

the membrane fraction, the supernatant was further centrifuged at $150,000 \times g$, 60 min at 4 °C. The resulting soluble supernatant (crude extract) was used to determine G3PDH activity. For preparation of the membrane fraction the pellet was washed in 5 mL of 50 mM MES-KOH, 20 mM KCl, pH 6.5, centrifuged ($150,000 \times g$, 60 min at 4 °C), resuspended in 500 μL of the same buffer and finally sonicated (20%, 0.5 s$^{-1}$, on ice) until membranes were fully suspended. This membrane fraction was then used for GK and G3PDH enzyme measurements. Enzyme activities in soluble and membrane fraction were assayed spectrophotometrically with protein amounts of 50 μg (crude extract) and 40 μg (membrane fraction) in a total volume of 500 μL (Specord UV/visible-light (Vis) spectrophotometer; Analytic Jena, Jena, Germany). Assay mixtures were preincubated at the respective temperatures before starting the reaction with substrate (glycerol or G3P). G3PDH activity was determined at 70 °C as G3P-dependent reduction of DCIPIP at 600 nm (extinction coefficient 20.7 mM$^{-1}$ cm$^{-1}$) in 50 mM MES-KOH pH 6.5, 20 mM KCl supplemented with 50 μM DCPIP and 200 μM G3P. GK activity was assayed at 50 °C in 100 mM TRIS-HCl pH 7, 1 mM MgCl$_2$, 1 mM ATP, 5 mM glycerol, 0.2 mM NADH, and 2 mM PEP by coupling the glycerol-dependent formation of ADP from ATP to NADH oxidation via PK (7 U) and LDH (14 U) at 340 nm (extinction coefficient 6.22 mM$^{-1}$ cm$^{-1}$). It was ensured that the auxiliary enzymes were not rate limiting. One unit (1 U) of enzyme activity is defined as 1 μmol substrate consumed or product formed per minute.

## Molecular cloning
*E. coli* DH5α, used for plasmid construction and propagation, was cultivated at 37 °C with shaking at 180 rpm (Unitron, INFORS HT, Bottmingen, Switzerland) in Lysogeny Broth (LB) supplemented with the appropriate antibiotics (150 μg mL$^{-1}$ ampicillin, 50 μg mL$^{-1}$ kanamycin, or 25 μg mL$^{-1}$ chloramphenicol). For heterologous overexpression in *E. coli* either the pET15b (*saci_2032, saci_1118*) or the pETDuet-1 vector (co-expresssion of *saci_2032/saci_2031* and *saci_1118/1119*, expression of *saci_2032* or *saci_1118* alone) and for homologous overexpression in *S. acidocaldarius* MW001 the pSVAmalFX-SH10 vector (*saci_1117*), pBS-Ara-albaUTR-FX-vector (*saci_2033*)[40], and the pSVAAraFX-HA vector (*saci_2032, saci_1119*)[73] were used. Plasmids and strains are listed in Supplementary Table 6. Open reading frames (ORFs) were amplified from genomic DNA of *S. acidocaldarius* DSM 639 and cloned into the respective vectors using the primers and restriction sites/endonucleases listed in Supplementary Table 7. For construction of pETDuet-1 co-expression vectors the gene pairs *saci_2032* and *saci_2031* or *saci_1118* and *saci_1119* were sequentially ligated into the multiple cloning site 1 (*saci_2032/saci_1118*) and 2 (*saci_2031/saci_1119*). Successful cloning was confirmed by sequencing.

## Construction of markerless single and double glycerol kinase deletion mutants Δsaci_1117, Δsaci_2033 and Δsaci_1117 Δsaci_2033
To obtain the markerless GK deletion mutants of *saci_1117* and *saci_2033*, the plasmids pSVA12818 and pSVA12822 were constructed. Briefly, 500 bp of the upstream and downstream region of each gene were amplified by PCR and the respective PCR products were annealed via overlap extension PCR. The resulting products were then cloned into pSVA407 or pSVA431 (for plasmids and strains, see Supplementary Table 6). The resulting plasmids were then methylated by transformation in *E. coli* ER1821 to prevent plasmid degradation in the recipient strain. Methylated plasmid was then used to transform MW00G as described previously for MW001[61]. To prepare competent MW00G, a pre-culture was grown in Brock media supplemented with 0.1% (w/v) N-Z-amine and 10 mM glycerol and 20 μg mL$^{-1}$ uracil to an OD$_{600}$ of 0.5–0.7 and used to inoculate 50 mL Brock media supplemented with 0.1% (w/v) N-Z-amine and 10 mM glycerol. Notably, competence could only be obtained after addition of glycerol to the culture medium. The culture was harvested at an OD$_{600}$ of 0.2–0.3 and further prepared as described previously[61].

## Complementation studies
For the complementation of the GK deletion mutants, MW1257 *Δ1117* and MW1258 *Δ2033*, *saci_1117* and the operon *saci_2033-2034* with 200 bp of the respective native promoter region and 500 bp of the *upsE* (*saci_1494*) upstream and downstream region were amplified using the primers listed in the Supplementary Table 7. The resulting PCR products, either P$_{saci\_1117}$ *saci_1117* or P$_{saci\_2033}$ *saci_2033-2034* combined with the *upsE* upstream and downstream PCR products, were then assembled in an *ApaI* and *SacII* digested pSVA407 vector via Gibson Assembly (New England Biolabs, Frankfurt a.M., Germany) following the manufacturer's instructions. The respective deletion mutants were then transformed with the resulting knock-in plasmids pSVA407-ΔupsE:P$_{1117}$ *saci_1117* CtSS and pSVA407-ΔupsE:P$_{2033}$ *saci_2033-2034*. Correct integration was confirmed by colony PCR and sequencing. Complementation was assessed by comparing growth and crude extract activities in Brock medium supplemented with 10 mM glycerol or 0.2% (w/v) of N-Z-amine, as described previously.

## Homologous overexpression and affinity purification of glycerol kinase
For homologous GK expression, the vectors pSVAmalFX-*saci_1117*-SH10 and pBS-Ara-albaUTR-FX-*saci_2033*-CtSS were methylated by transforming them into *E. coli* ER1821, thereby preventing degradation in the recipient strain. The methylated plasmids were transformed into *S. acidocaldarius* MW001 via electroporation using a Gene Pulser Xcell (BioRad, Munich, Germany) with a constant time protocol (input parameters 2 kV, 25 μF, 600 Ω in 0.5 mm cuvettes). Cells were incubated for 45 min at 75 °C in recovery solution[61], and then plated on Brock's basal salt medium with 0.1% (w/v) N-Z-amine, 0.2% (w/v) dextrin solidified by gelrite without uracil. Successful transformation was confirmed via colony-PCR. Clones were precultured in 50 mL liquid Brock's basal salt medium supplemented with 0.1% (w/v) N-Z-amine, 0.2% (w/v) dextrin and then used to inoculate 2 L liquid Brock's basal salt medium (OD$_{600}$ 0.05) containing 0.1% (w/v) N-Z-amine and 0.2% (w/v) D-xylose for induction. Cells were grown for two days at 76 °C under constant shaking at 120 rpm, harvested by centrifugation at $7000 \times g$ and 4 °C for 15 min and stored at −80 °C for further use. For protein purification, the cells were resuspended in 50 mM TRIS-HCl buffer (pH 8.0), 250 mM NaCl at a ratio of 3 mL buffer per 1 g of wet cell weight. The cells were then disrupted by sonication (3 × 8 min, 60% amplitude, 0.5 s$^{-1}$, on ice). Cell debris was removed by centrifugation at $30,000 \times g$ and 4 °C for 45 min and the resulting crude extract was applied onto a Strep-Tactin®XT 4Flow® column (IBA Lifesciences, Göttingen, Germany) and purified following the manufacturers' instructions.

## Heterologous overexpression and affinity purification of G3PDHs
For the expression of G3PDHs, *E. coli* Rosetta (DE3) cells transformed with the vectors pET15b-*saci_2032* and pET15b-*saci_1118*, respectively, were cultured in terrific broth (TB) supplemented with 150 μg mL$^{-1}$ ampicillin and 25 μg mL$^{-1}$ chloramphenicol at 37 °C. When the culture reached an OD$_{600}$ of 0.8, expression was induced with 1 mM IPTG, followed by further growth at 20 °C for 17 hours. Cells were harvested by centrifugation ($6000 \times g$, 20 min, and 4 °C) and stored at −80 °C for further use. Cells were resuspended in 10 mL of buffer (50 mM TRIS-HCl pH 7.8, 10 mM imidazole, 150 mM NaCl, and 1 mM FAD) per gram of cells (wet weight) and disrupted by sonication (3 × 8 min, 50% amplitude, 0.5 s$^{-1}$, on ice). The cell debris was removed by centrifugation at $16,000 \times g$ for 45 min at 4 °C. After filtration (0.45 μm polyvinylidene fluoride membrane; Starlab, Hamburg, Germany), the supernatant was applied onto a 1 mL Nickel-IDA column (Cytiva, Marlborough, MA, USA) equilibrated in 50 mM TRIS-HCl pH 7.8, 10 mM imidazole, 150 mM NaCl and after washing (20 column volumes) proteins were eluted with a linear gradient from 10–300 mM imidazole in equilibration buffer using an ÄktaPurifier system (Cytiva, Marlborough, MA, USA). In G3PDH containing fractions imidazole was removed using a centrifugal concentrator (10 kDa cutoff, Sartorius AG, Göttingen, Germany). Protein aliquots were frozen in liquid nitrogen and stored at −80 °C for further use.

### Determination of the native molecular mass of purified proteins

Size exclusion chromatography (SEC) was used to determine the native molecular mass of all purified enzymes. To this end, pooled enzyme samples after affinity chromatography were concentrated using centrifugal concentrators (10 kDa cutoff) and applied to a Superose 6 Increase 10/300 GL column (Cytiva, Marlborough, MA, USA). For both GKs, Saci_1117 and Saci_2033, 50 μg protein in 200 μL were applied. For G3PDHs, 19 mg Saci_2032 or 6 mg Saci_1118 each in 500 μL were loaded to the column. 50 mM TRIS-HCl pH 7.5, 250 mM NaCl was employed as running buffer. For calibration, the LMW and HMW Gel Filtration Calibration Kits (Cytiva, Marlborough, MA, USA) were used.

### Localization of G3PDH upon co-expression with CoxG in *E. coli*

To investigate the role of CoxG (Saci_2031 and Saci_1119) for membrane anchoring of the G3PDHs (Saci_2032 and Saci_1118) in *S. acidocaldarius*, co-expression studies in *E. coli* were performed. Therefore, *saci_2032* and *saci_2031* or *saci_1118* and *saci_1119* were co-expressed from pETDuet-1 expression vectors in *E. coli* Rosetta (DE3) as described above. As a control, *saci_2032* or *saci_1118* were expressed alone from the same vector without *saci_2031* or *saci_1119*. After expression, cells were disrupted as described above and the resulting supernatant was centrifuged at $150,000 \times g$ at 4 °C for 120 min. The soluble fraction (supernatant) was separated from the membrane fraction, which was washed twice with 50 mM TRIS-HCl, pH 7.0 and centrifuged ($150,000 \times g$, 4 °C, 120 min). Finally, the membrane fraction and soluble fraction were aliquoted, flash frozen in liquid nitrogen, and stored at −80 °C. Isolated membranes were resuspended in 500 μL of buffer (50 mM TRIS-HCl, pH 7.0) and sonicated at low amplitude (20%, 0.5 s⁻¹, on ice) until fully homogenized. 50 μg of the membrane or soluble protein fractions were subjected to SDS-PAGE and transferred to a PVDF membrane (Roth, Karlsruhe, Germany). Immunodetection was performed with horseradish peroxidase (HRP) conjugated anti-His antibody (1:10,000 dilution; Abcam, Cambridge, United Kingdom) and the ClarityTM Western ECL substrate (BioRad, Munich, Germany) using a VersaDoc (BioRad) or Fusion FX (Vilber Lourmat, Marne-la-Vallée, France) imaging system.

### Localization of CoxG homologues in vivo and co-immunoprecipitation (co-IP) assays with anti-HA-magnetic beads

S*aci_2031* or *saci_1119* were homologously expressed from the pSVAaraFX-HA vector construct transformed into *S. acidocaldarius* MW00G to yield a C-terminally HA-tagged recombinant protein. Expression cultures were grown in Brock's basal salt medium with 10 mM glycerol as carbon source for 72 h to an $OD_{600}$ 0.6–0.7 and expression was induced by the addition of 0.2% (w/v) L-arabinose. After 4 h of additional growth ($OD_{600}$ 0.8–0.9), the cells were cooled and harvested by centrifugation at $5000 \times g$ for 20 min at 4 °C. For localization of the CoxG homologues, cells were disrupted and soluble and membrane fractions were separated as described above for co-expression studies in *E. coli*. HA-tagged CoxG homologues in the separated fractions were detected by western blotting and immunodetection using anti-HA and horse radish peroxidase-coupled secondary antibodies (Invitrogen, Thermo Fisher Scientific, Waltham, Massachusetts, United States). For Co-IP/pull-down experiments, pellets obtained from 500 mL expression cultures were resuspended to a calculated $OD_{600}$ of 40 in 25 mM TRIS-HCl pH 7, 150 mM KCl, 5% (v/v) glycerol, 10 mM EDTA, and 2% (w/v) n-dodecyl β-maltoside (DDM) and incubated for 1 h at 37 °C while rotating. Samples were centrifuged (30 min, $15,000 \times g$, room temperature (RT)) to remove cell debris and the supernatant was used for co-immunoprecipitation. Therefore, 35 μL of Pierce™ anti-HA Magnetic Beads (Pierce™ HA-tag IP/co-IP kit (Pierce, Thermo Fisher Scientific)) were added to 10 mL of the cell lysate and incubated for 1 h at room temperature while rotating. The beads were collected on a magnetic stand and washed twice with 25 mM TRIS-HCl pH 7 containing 150 mM KCl, 5% (v/v) glycerol, 10 mM EDTA and finally once with LC-MS grade water (Merck, Darmstadt, Germany). After washing, 35 μL of LC-MS grade water was

added and 25 μL bead suspension was used for on-bead digestion as described previously[67].

### Characterization of purified enzymes

Enzyme assays were performed spectrophotometrically using a Specord UV/visible-light (Vis) spectrophotometer in a total volume of 500 μL (unless stated otherwise). Assay mixtures were pre-warmed to the designated assay temperature, and the reaction was typically initiated by the addition of the substrate. Experimental data were fitted, and kinetic constants were determined using the OriginPro 2022 software package (OriginLab).

**Glycerol kinase.** GK activity was determined as described above for the crude extract measurements with the following modifications: For the continuous PK-LDH assay at 50 °C, the assays were performed for Saci_1117 in 0.1 M TRIS-HCl buffer (pH 7 at 50 °C), 1 mM MgCl₂, 1 mM ATP, 0.2 mM NADH, 2 mM PEP, 14 U LDH, 7 U PK using 0.5 μg of purified Saci_1117. For Saci_2033 the assay was conducted in 0.1 M MOPS-KOH buffer (pH 6.5 at 50 °C), 5 mM MgCl₂, 5 mM ATP, 0.2 mM NADH, 2 mM PEP, 14 U LDH, 7 U PK using 0.25 μg of purified Saci_2033. To determine the kinetic properties varying concentrations of glycerol (0–2 mM), DHA (0–30 mM), DL-glyceraldehyde (0–20 mM) or ATP (0-5 mM) were used at a constant concentration of 1 mM (Saci_1117) and 5 mM (Saci_2033) ATP, 1 mM (Saci_1117) and 5 mM (Saci_2033) MgCl₂ or 2 mM glycerol, respectively. One unit (1 U) of enzyme activity is defined as 1 μmol of product (ADP) formed per minute. Substrate specificity was tested using the continuous PK-LDH assay at 50 °C with 5 mM of various substrates, including DL-glyceraldehyde (GA), dihydroxyacetone (DHA), DL-glyceric acid, D-xylose, D-glucose, meso-erythritol, D-sorbitol, or xylitol, in place of glycerol. Nucleotide specificity was assessed using the G3PDH-coupled assay, as described below, by substituting 5 mM GTP, CTP, or phosphoenolpyruvate (PEP) for ATP at 75 °C.

The pH optimum was determined using the PK-LDH assay at 50 °C in a mixed buffer system ranging from pH 5.0 to pH 8.0 containing 50 mM MES, 50 mM HEPES and 50 mM TRIS-HCl. The temperature optimum was determined between 60 °C to 80 °C using the continuous G3PDH assay. Therefore, the glycerol kinase (GK)-mediated glycerol-3-phosphate (G3P) formation was coupled to DCPIP reduction, monitored at 600 nm, using purified recombinant G3PDH (0.25 U) from *Sulfolobus acidocaldarius* (Saci_2032). The assay for Saci_1117 was conducted in 0.1 M TRIS-HCl buffer (pH 7, temperature adjusted), containing 1 mM ATP, 1 mM MgCl₂, 0.1 mM DCPIP, and 2 mM glycerol. For Saci_2033, the assay was performed in 0.1 M MOPS-KOH buffer (pH 6.5, temperature adjusted), with 5 mM ATP, 5 mM MgCl₂, 0.1 mM DCPIP, and 2 mM glycerol. To study the thermal stability, the GKs Saci_1117 and Saci_2033 were incubated at 70 °C, 80 °C, and 90 °C in 150 μL (total volume) 100 mM TRIS-HCl (pH 7) or MOPS-KOH (pH 6.5) (pH temperature adjusted), respectively, at a protein concentration of 0.05 mg mL⁻¹. At regular time intervals (1 h, 3 h, 6 h, and 24 h), samples (10 μL for Saci_1117 and 5 μL for Saci_2033) were taken, and the residual enzyme activity was measured using the continuous PK-LDH assay at 50 °C, as previously described. The effect of fructose 1,6-bisphosphate (F1,6BP) on GK activity was also evaluated using the continuous PK-LDH assay (described above) with F1,6BP concentrations of up to 1 mM, as indicated.

**Glycerol-3-phosphate dehydrogenase.** The determination of the reduction of enzyme-bound FAD was performed at 70 °C (200 μL total volume) in 50 mM HEPES-KOH pH 6.5 (temperature adjusted), 100 mM KCl, 50 μM of G3P and 70 μg of Saci_2032 or Saci_1118, corresponding to a final concentration 5 μM. The reaction was started by addition of G3P and after 5 min absorption spectra were measured between 400 and 500 nm in 96 well plates in a Tecan Infinite M200 plate reader. Loss of absorption at 450 nm indicates the reduction of bound FAD. To determine the FAD concentration of the purified G3PDH, the

protein was denatured in 50 mM HEPES-KOH pH 6.5, 100 mM KCl with 0.5% (w/v) SDS (500 μL total volume) at room temperature and FAD absorbance spectra were determined from 300 nm to 600 nm (Specord UV/visible-light (Vis) spectrophotometer). The FAD content was calculated using the extinction coefficient of 11.300 mM$^{-1}$ cm$^{-1}$ at 450 nm.

To test the quinone reactivity, 0.6 μg of Saci_2032 or 1.2 μg of Saci_1118 were preincubated with 30 μM of water-soluble ubiquinone-Q1 (500 μL total volume) in 50 mM HEPES-KOH pH 6.5, 100 mM KCl. Native ubiquinones used in Bacteria and eukaryotes are water-insoluble due to side chains comprising up to ten isoprenoid units. Afterwards 100 μM of G3P was added to the samples and reduction of ubiquinone-Q1 was followed in 15 s intervals by recording the absorption spectra between 260 and 330 nm using a Specord UV/visible-light (Vis) spectrophotometer. Loss of absorption at 280 nm indicates the reduction of the ubiquinone-Q1.

For enzyme characterization continuous enzyme assays were conducted at 70 °C (500 μL total volume) by monitoring the decrease in absorbance during the reduction of the artificial electron acceptor DCPIP at 600 nm or ubiquinone-Q1 at 280 nm. Kinetic parameters for G3P were determined in 50 mM HEPES-KOH pH 6.5, 100 mM KCl, 0.06 mM DCPIP, with 0.35 μg of Saci_2032 or 0.5 ug of Saci_1118 at 70 °C and varying concentrations of G3P (0–0.3 mM). To determine the kinetic constants for ubiquinone-Q1, DCPIP was omitted and the concentration of the quinone was varied between 0 mM and 0.06 mM with a concentration of 0.6 μg for either Saci_2032 or Saci_1118, while maintaining a fixed G3P concentration of 0.4 mM. One unit (1 U) of enzyme activity is defined as 1 μmol of product (DCPIP$_{red}$ or ubiquinone-Q1$_{red}$) formed per minute. Substrate specificity was evaluated by replacing glycerol with 0.3 mM D-glycerol 1-phosphate, D-glycerol, DL-glyceric acid, D-glyceraldehyde 3-phosphate, and D-phosphoglyceric acid.

Both, the pH and the temperature optimum were determined using 0.4 mM G3P and 0.06 mM DCPIP. For the pH optimum a mixed buffer system containing 50 mM MES, 50 mM HEPES, and 50 mM TRIS was used in the range of pH 5.0 to pH 8.0. The temperature optimum between 50 °C and 80 °C was determined in 50 mM MES-KOH pH 6.5 adjusted at the respective temperature. Glycerol oxidase activity of both G3PDHs was tested as G3P-dependent hydrogen peroxide formation coupled to the oxidation of 2,2'-azinobis-(3-ethylbenzothiazoline-6-sulfonate) (ABTS) via horseradish peroxidase (HPR, Merck, Darmstadt, Germany) measured as increase in absorbance at 420 nm (extinction coefficient 42.3 mM$^{-1}$ cm$^{-1}$)[25]. The assays were performed in 100 mM MES-KOH pH 6.5, 1 mM ABTS, and 0.2 U of HRP with 0.045 μg of protein. The thermostability of the Saci_2032 and Saci_1118 G3PDHs was analyzed by incubating the enzyme at 70 °C, 80 °C, and 90 °C in 50 mM MES-KOH, pH 6.5 (temperature adjusted, 400 μL total volume) at a protein concentration of 0.18 μg μL$^{-1}$. After regular time intervals (1, 3, and 6 h) 10 μL aliquots were removed and the residual activity of the enzyme was determined at 70 °C with G3P and DCPIP as described above. The influence of detergents and membrane lipids on Saci_2032 and Saci_1118 activity was analysed with G3P and DCPIP at 50 °C in the presence of either 0.5% (w/v) DDM, 0.5% (v/v) of triton X-100, 50 μg of phosphatidylcholine (Merck, Darmstadt, Germany) or 50 μg of isolated S. acidocaldarius MW00G membrane fractions (prepared as described above).

## Analytical assays
If not stated otherwise, protein purity was analyzed by SDS-PAGE and protein concentration was determined by a modified Bradford assay[74] using bovine serum albumin (Carl Roth, Germany) as standard.

## Bioinformatic analyses
Structural models were retrieved from the AlphaFold Protein Structure database[75,76] or predicted using the ColabFold software[77]. Structural analyses, comparisons, and visualizations were done using UCSF Chimera package from the Resource for Biocomputing, Visualization, and Informatics at the University of California, San Francisco (supported by NIH P41

RR-01081)[78]. For phylogenetic analyses sequences were aligned with clustal omega using the EMBL server[79]. The clustal omega alignment was used in the MEGA11 software package for phylogenetic tree constructions[80] (for further details see legend to Fig. 8).

## Statistics and Reproducibility
All data was generated in three independent experiments (n = 3), with the exception of the metabolite analysis that was preformed using eight replicates (n = 8). Sample sizes were chosen based on standard sample sizes in literature. All graphs were created using OriginPro2024. All statistical details are described in the figure legends or in the methods section of the corresponding experiment.

## Reporting summary
Further information on research design is available in the Nature Portfolio Reporting Summary linked to this article.

## Data availability
The mass spectrometry proteomics data for the on-bead digestions have been deposited to the ProteomeXchange Consortium via the PRIDE[81] partner repository (https://www.ebi.ac.uk/pride/archive/) with the dataset identifier PXD050086. All other data generated during the current study are available from the corresponding author upon request. RNA-seq data were deposited in in the ArrayExpress database (www.ebi.ac.uk/arrayexpress) under accession number E-MTAB-14293. Numerical source data for all graphs can be found in Supplementary File 4 and unedited versions of gel pictures and western blots are shown in Supplementary Fig. 14.

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

## Acknowledgements

This research was funded by the Federal Ministry of Education and Research (BMBF) and by the Volkswagen Foundation. B.S., M.K., O.J.S. (031B0848A), J.K. (031B0848B) and S-V.A. (031B0848C) wish to express their gratitude for the financial support received for the HotAcidFACTORY project, titled 'Sulfolobus acidocaldarius as a novel thermoacidophilic bio-factory', under the BMBF National Bioeconomy Strategy (Microbial Biofactories for Industrial Bioeconomy—Novel Platform Organisms for Innovative Products and Sustainable Bioprocesses). B.S. and C.B. wish to express their gratitude for the financial support received for the Lipid ∥ Divide—'Resolving the 'lipid divide' by unravelling the evolution and role of fatty acid metabolic pathways in Archaea' project within the 'Life?'—A fresh scientific approach to the basic principles of life initiative (Volkswagen Foundation, grant number 96725). We acknowledge support by the Open Access Publication Fund of the University of Duisburg-Essen.

## Author contributions

Ch.S., X.Z. and Ca.S. conducted growth studies, crude extract measurements, cloning and expression, as well as enzyme purification and characterization. T.B. performed RNA sequencing and related data analysis. S.N. and F.K. executed proteomics and corresponding data analysis. L.M. carried out metabolomics and associated data analyses. J.B. and B.W. constructed the GK deletion mutants, cloned and homologously expressed CoxG, and conducted co-immunoprecipitation experiments. C.B. handled sequence, structural, and phylogenetic analyses. The study was conceived and designed by S-V.A., J.K., M.K., O.J.S., C.B. and B.S. All authors contributed to writing the initial draft, while Ch.S., Ca.S., C.B. and B.S. reviewed and edited the final draft. Project administration was managed by B.S., and funding acquisition was secured by J.K., S-V.A., O.J.S., M.K., C.B. and B.S. All authors contributed to the article and approved the final submitted version.

## Funding

## Competing interests
The authors declare no competing interests.
