## [Transparent Peer Review file · Communications Biology]

The identification of an unusual glycerol-3-phosphate dehydrogenase in *Sulfolobus acidocaldarius* elucidates the evolution and functional diversity of glycerol metabolism enzymes across Archaea

Corresponding Author: Professor Bettina Siebers

This manuscript has been previously reviewed at another journal. This document only contains information relating to versions considered at Communications Biology.

Version 0:

Reviewer comments:

Reviewer #1

(Remarks to the Author)

Schmerling et al. characterized glycerol utilization and the enzyme glycerol 3-phosphate dehydrogenase from *Sulfolobus acidocaldarius*. While the overall sequence of reactions, facilitator, kinase and dehydrogenase is the same as in *Escherichia coli*, the enzyme glycerol-3-phosphate dehydrogenase (G3PDH) is unusual as it exhibits a previously undescribed type of membrane anchoring via a CoxG-like protein. The study does not stop with omics data, but provides excellent data from enzyme characterization to achieve a mechanistic understanding. The study also provides genetic evidence for the role of glycerol kinases.

The study is carried out well and comprehensively, addresses a very relevant subject with a particular focus on archaea. An evolutionary perspective is provided.

Major points:

Is there interaction between G3PDH and CoxG homologues from the two operons `saci_1118/1119` and `saci_2032/2031`?

Are none/one or both CoxG homologues required for glycerol utilization?

Are none/one or both G3PDH homologues essential for growth with glycerol ?

Minor points:

Results section and supplement: Please calculate biomass yields for glycerol and xylose in g/g as well

Results section: does glycerol downregulate Weimberg pathway for pentose degradation genes/proteins or does xylose upregulate these?

Results section and supplement: please provide error estimates, e.g., as std. dev. for K_M , V_{max} values.

Reviewer #2

(Remarks to the Author)

This is the best paper I reviewed in the last year.

The authors reveal the glycerol metabolism in the hyperthermophilic archaeon *Sulfolobus acidocaldarius*, from physiological (growth) studies via identification of the major involved proteins in glycerol catabolism by omics, deletion of the respective genes and purification of the overproduced enzymes for functional characterization. Lastly, the unusual glycerol-3-p-DH was found to form a distinct phylogenetic cluster.

The paper is clearly-written and easy to follow, the experimental approach is straight-forward, the results are well presented and discussed. It is also important since there was only one, very different example from the domain archaea where glycerol metabolism had been studied before.

I only have minor comments.

Fig. 9. : Does the haloarchaeal GlpA-type G3PDH look similar to the E. coli one?

Fig. 10 : What is the membrane-integral electron carrier for GlpABC - the isoprenoid quinone caldariellaquinone, as suggested in Fig. 10? Is this or a water-soluble homologue available for biochemical enzymes assays, and would you assume a different function of CoxG on activity here? Is ubiquinone or the artificial redox active dye DCPIP the regularly used analogon for such enzyme assays? Except for the functional group, it is very different in structure and chain length.

Luckily, I have no comment referring to a line number. Please include line number next time. Potentially, this may not be asked for by the journal, but it simplifies the review process.

Great work!

Version 1:

Reviewer comments:

Reviewer #1

(Remarks to the Author)

I am satisfied with the responses even by the authors.

Reviewer #2

(Remarks to the Author)

My comments were addressed very well. In my opinion, the manuscript may now be published as is.

Reply to the reviewer comments

First, we would like to thank the reviewers for their efforts in evaluating our manuscript and for their valuable and constructive comments, which we hope have been satisfactorily addressed, thereby improving the paper considerably. Please find below our detailed responses to the points raised in your review.

Dear Professor Siebers,

Your manuscript entitled "The identification of an unusual glycerol-3-phosphate dehydrogenase in *Sulfolobus acidocaldarius* elucidates the evolution and functional diversity of glycerol metabolism enzymes across Archaea" has now been seen by 2 referees. You will see from their comments below that while they find your work of considerable interest, some important points are raised. We are interested in the possibility of publishing your study in *Communications Biology*, but would like to consider your response to these concerns in the form of a revised manuscript before we make a final decision on publication.

We therefore invite you to revise and resubmit your manuscript, taking into account the points raised. In particular,

* clarify a bit better on the physiological need of the two G3PDH homologs e.g. for glycerol utilisation and include a bit more on the similarities to other e.g. bacterial G3PDH. Finally, comment a bit more on the quinone topic as requested by R2.

As mentioned in the discussion section two G3PDH are present in several organisms, but the physiological function of two enzymes in parallel is only well established for *E. coli* where GlpD and GlpABC are used under different conditions, i.e. during aerobic and anaerobic growth, respectively. Using deletion mutant analyses, we herein showed that *saci_2031-34* is mainly involved in breakdown of free glycerol (also supported by the presence of the GUF glycerol transporter (*Saci_2234*) in the gene cluster) whereas the experiments did not reveal a clear function of the *saci_1117-1119* since this gene cluster appears dispensable and not strongly regulated under the tested conditions. However, the genomic context (*saci_1103-1126*) of this second glycerol gene clusters *saci_1117-1119* in *S. acidocaldarius* comprises genes likely involved in lipid hydrolysis and fatty acid β oxidation which might suggest a function in glycerol utilization during lipid degradation. On solid media containing triglycerides like tributyrine, *S. acidocaldarius* was shown to be active and able to cleave these lipids by means of esterases encoded in the *saci_1103-1126* gene neighborhood (Zweerink et al., 2017, DOI: 10.1038/ncomms15352). We added a short comment on that in the discussion (lines 434-440).

The editor's request to add on similarities to other e.g. bacterial G3PDHs presumably refers to the comment of reviewer 2 concerning the similarities of bacterial and haloarchaeal G3PDHs (please see our answer below). Finally, we made some additions to the text (introduction, results, discussion and materials and methods) to clarify the quinone topic (for details also see comments below), and hope that we addressed the editor's as well as the reviewer requests and comments satisfactorily.

Please highlight all changes in the manuscript text file.

Done (track change function in MS word). We also added line numbers (as also requested by reviewer 2) to make it easier to follow the changes made to the manuscript.

We also had to add three new references (reference numbers 43, 54, 79; highlighted in cyan in the reference list and in text citations).

To Table 1 (main text) as well as to the supplementary tables 3 and 4 the following footnote was added: "Parameter values (obtained in this work) are given \pm standard deviation."

To supplementary table 1 the footnote "Means of $n \geq 3$ biological replicates are shown \pm standard deviation." was added.

We are committed to providing a fair and constructive peer-review process. Do not hesitate to contact us if you wish to discuss the revision in more detail or if there are specific requests from the reviewers that you believe are technically impossible or unlikely to yield a meaningful outcome.

At the same time, we ask that you ensure your manuscript complies with our editorial policies. Please see our revision file checklist for guidance on formatting the manuscript and complying with our policies. You will also find guidelines for replying to the referees' comments. You may also wish to review our formatting guidelines for final submissions here.

Please use the following link to submit your revised manuscript, point-by-point response to the referees' comments (which should be in a separate document to the cover letter) and any additional files:

When submitting the revised version of your manuscript, please pay close attention to our Digital Image Integrity Guidelines.

We would like to receive your revision within 4 weeks, but appreciate that every situation is unique. We look forward to receiving your revised manuscript when it is ready, and will not enforce a hard deadline on this revision.

Please do not hesitate to contact me if you have any questions or would like to discuss these revisions further. We look forward to seeing the revised manuscript and thank you for the opportunity to review your work.

Best regards and a nice weekend,
Tobias Goris, PhD
Senior Editor
Communications Biology
orcid.org/0000-0002-9977-5994

On behalf of

Kaliya Georgieva, PhD
Associate Editor
Communications Biology
<https://orcid.org/0009-0006-2251-2578>

Referee expertise:

Referee #1: Prokaryotic Genetics and Physiology

Referee #2: archaeal/bacterial microbiology and biotech applications

Reviewers' comments:

First, we would like to thank the reviewers for their efforts in evaluating our manuscript and for their valuable and constructive comments which we hope have been satisfactorily addressed, thereby improving the paper considerably.

Reviewer #1 (Remarks to the Author):

Schmerling et al. characterized glycerol utilization and the enzyme glycerol 3-phosphate dehydrogenase from *Sulfolobus acidocaldarius*. While the overall sequence of reactions, facilitator, kinase and dehydrogenase is the same as in *Escherichia coli*, the enzyme glycerol-3-phosphate dehydrogenase (G3PDH) is unusual as it exhibits a previously undescribed type of membrane anchoring via a CoxG-like protein. The study does not stop with omics data, but provides excellent data from enzyme characterization to achieve a mechanistic understanding. The study also provides genetic evidence for the role of glycerol kinases.

The study is carried out well and comprehensively, addresses a very relevant subject with a particular focus on archaea. An evolutionary perspective is provided.

Major points:

Is there interaction between G3PDH and CoxG homologues from the two operons *saci_1118/1119* and *saci_2032/2031*?

The co-immunoprecipitation experiments (Supplementary Table 5) indicate that at least *Saci_2031* CoxG interact not only with *Saci_2032* G3PDH but to some extent also with *Saci_1118* G3PDH. So, there might be a certain degree of "cross-interaction" between *Saci_2032/31* and *Saci_1118/19*. However, the CoxGs alone resided in the (insoluble) membrane fraction so that biochemical assays to further elucidate this e.g. via *in vitro* reconstitution experiments with the purified subunits were impossible. We therefore decided, not to discuss these findings further for clarity and readability of the manuscript (see also further comments below).

Are none/one or both CoxG homologues required for glycerol utilization?

Are none/one or both G3PDH homologues essential for growth with glycerol ?

To address these questions we initially aimed to construct the respective single and in case also double deletion mutants of the *g3pdh* and *coxG* genes in addition to the *gk* genes. However, we faced significant challenges in establishing any mutants in the glycerol adapted *S. acidocaldarius* strain (mutant constructions were performed in the laboratory of Sonja-Verena Albers) and it was only after adapting the mutant selection strategy to include 10 mM glycerol that we successfully generated mutants. Since this whole process was extremely time intensive, we decided to prioritize the understanding which of the two gene clusters plays the key role in the glycerol degradation and focused on the glycerol kinases which catalyze the first step in the breakdown. The deletion of the GK in contrast to the G3PDH avoids the accumulation of glycerol-3-phosphate, which was shown to interfere with regulation and impede growth on glycerol in *Pseudomonas* spp. (Liu et al., 2022 DOI: [10.1128/mbio.02624-22](https://doi.org/10.1128/mbio.02624-22); Poblete-Castro et. Al., 2020 DOI: <https://doi.org/10.1111/1751-7915.13400>) and thus might have further impaired the mutant construction in *S. acidocaldarius*.

Thus, the generation of the remaining mutants and their combinations as well as the analyses of the cross-interaction and function of the G3PDH holoenzymes (e.g. via coexpression analyses of the G3PDH and CoxGs in different combination) represent distinct research projects that we plan to pursue in the future.

Minor points:

Results section and supplement: Please calculate biomass yields for glycerol and xylose in g/g as well

Done, we added the requested information and included an additional column in supplementary Table 1 and in the results “higher biomass yield on glycerol” (line 150-151). Furthermore, we added the standard deviation for the growth rates (main text line 144, 149). In supplementary Table 1, we also added the standard deviations for growth rates, CDW, and (molar) growth yields.

Results section: does glycerol down regulate Weimberg pathway for pentose degradation genes/proteins or does xylose upregulate these?

Our previous study (Wagner et al., 2017) showed that the Weimberg pathway is up-regulated in the presence of D-xylose. This finding was confirmed in this study (Supplementary Table 2), which demonstrated that the Weimberg pathway is down-regulated in the absence of D-xylose, when *S. acidocaldarius* is grown on glycerol minimal medium or NZ-amine alone. We added a short comment in the results (since it does hardly fit into the text flux in the discussion) (line 158-159, and newly included the reference 43 (Wagner et al., 2017, see above).

Results section and supplement: please provide error estimates, e.g., as std. dev. for KM, Vmax values.

Done, we added the requested information in the main text where appropriate (Results section, lines, 168, 169, 173, 200-203, 220-221, 280) and to Table 1 as well as Supplementary Tables 3 and 4.

Reviewer #2 (Remarks to the Author):

This is the best paper I reviewed in the last year.

The authors reveal the glycerol metabolism in the hyperthermophilic archaeon *Sulfolobus acidocaldarius*, from physiological (growth) studies via identification of the major involved proteins in glycerol catabolism by omics, deletion of the respective genes and purification of the overproduced enzymes for functional characterization. Lastly, the unusual glycerol-3-p-DH was found to form a distinct phylogenetic cluster.

The paper is clearly-written and easy to follow, the experimental approach is straight-forward, the results are well presented and discussed. It is also important since there was only one, very different example from the domain archaea where glycerol metabolism had been studied before.

I only have minor comments.

Fig. 9. : Does the haloarchaeal GlpA-type G3PDH look similar to the *E. coli* one?

The alphafold model of *H. volcanii* G3PDH superimposes well with the GlpA model from *E. coli*. However, the halophilic GlpAs have a distinctive C-terminal extension of approximately 30 amino acids. Interestingly, *H. volcanii* GlpC also exhibits an N-terminal extension, which may suggest a structural

role in complex formation. Unfortunately, alphafold failed to predict secondary structure or fold for these extensions, making it challenging to identify similar folds and assign functions. To maintain clarity and avoid confusion, we have omitted these findings from the discussion but included a comment in the Supplementary information (page 7).

Fig. 10 : What is the membrane-integral electron carrier for GlpABC - the isoprenoid quinone caldariellaquinone, as suggested in Fig. 10?

The electron acceptor for the canonical GlpABC G3PDH in bacteria, such as *E. coli*, is a menaquinone (as shown in Figure 1, which illustrates glycerol degradation pathways in various bacteria and eukaryotes under aerobic and/or anaerobic conditions). Typically, bacteria use ubiquinone under aerobic conditions and menaquinone under anaerobic conditions. Facultative organisms like *E. coli* switch between these quinones in response to oxygen availability. Thus, menaquinone serves as the electron acceptor for the "anaerobic" GlpABC, while ubiquinone is used by the "aerobic" GlpD. We made an addition to the introduction (lines 85-86 and 88-90) to explain this more clearly.

The caldariellaquinone is a unique quinone found only in members of the order Sulfolobales, including *S. acidocaldarius*, and serves as the primary quinone in these organisms. As shown in Fig. 10, it is likely that the caldariellaquinone functions as the electron acceptor for Saci_2032/2031 (GlpA/CoxG). We have added a sentence to the discussion with a reference to Fig. 10 to clarify this (lines 413-416).

Is this or a water-soluble homologue available for biochemical enzymes assays, and would you assume a different function of CoxG on activity here?

We attempted to use the purified caldariellaquinone extracted from *S. acidocaldarius* as the electron acceptor in the enzyme assay. However, its water insolubility made it completely unsuitable for use in aqueous enzyme assays. Unfortunately, a water-soluble analogue of the caldariellaquinone is to our knowledge not commercially available. Hence, we cannot rule out the possibility that the kinetic efficiency of the enzyme might be different when using the native substrate (analogue). Additionally, we cannot entirely exclude the possibility that the CoxG subunit influences G3PDH activity when using the native substrate. Nevertheless, our data strongly indicate the quinone reactivity of the GlpA subunit and that CoxG is not essential for enzyme activity, but rather for membrane association, and this is likely to be the case even when using the native substrate.

Is ubiquinone or the artificial redox active dye DCPIP the regularly used analogon for such enzyme assays? Except for the functional group, it is very different in structure and chain length.

The native caldariellaquinones, ubiquinones, and menaquinones used in respiratory chains are water-insoluble due to their long isoprenoid side chains (6-10 isoprenoid units), making them unsuitable for use in aqueous biochemical assays. To overcome this limitation, more soluble electron acceptors are often used, such as DCPIP (2,6-dichlorophenolindophenol), a redox-active dye that reacts directly with reduced FAD. This reaction is independent of the enzyme, as the electron transfer occurs directly from FAD to DCPIP. In contrast, ubiquinone-Q1 is a soluble analogue of ubiquinones, with a shortened side chain comprising only one isoprenoid unit. The transfer of electrons from FAD to ubiquinone-Q1 is enzyme-dependent, which allows to assess the ability of an enzyme to transfer electrons to quinones. Using ubiquinone-Q1 we demonstrate the quinone reactivity of the *S. acidocaldarius* G3PDHs. To clarify the quinone topic, we made a couple of additions to the text (Results: lines 224-229; Materials&Methods: lines 894-896). However, as previously noted, we cannot rule out the possibility that different kinetic properties may be observed with the native substrate caldariellaquinone.

Luckily, I have no comment referring to a line number. Please include line number next time. Potentially, this may not be asked for by the journal, but it simplifies the review process.

See above. We will definitely consider this for our future (initial) submissions.

Great work!

Thanks 😊 !

Bettina Siebers on behalf of all co-authors.